# FROM VARIATIONAL TO DETERMINISTIC AUTOENCODERS

**Partha Ghosh**†*     **Mehdi S. M. Sajjadi**†*     **Antonio Vergari**‡     **Michael Black**†

**Bernhard Schölkopf**†

† Max Planck Institute for Intelligent Systems, Tübingen, Germany
`{pghosh,msajjadi,black,bs}@tue.mpg.de`
‡ University of California, Los Angeles, USA
`aver@cs.ucla.edu`

## ABSTRACT

Variational Autoencoders (VAEs) provide a theoretically-backed and popular framework for deep generative models. However, learning a VAE from data poses still unanswered theoretical questions and considerable practical challenges. In this work, we propose an alternative framework for generative modeling that is simpler, easier to train, and deterministic, yet has many of the advantages of VAEs. We observe that sampling a stochastic encoder in a Gaussian VAE can be interpreted as simply injecting noise into the input of a deterministic decoder. We investigate how substituting this kind of stochasticity, with other explicit and implicit regularization schemes, can lead to an equally smooth and meaningful latent space without forcing it to conform to an arbitrarily chosen prior. To retrieve a generative mechanism to sample new data, we introduce an ex-post density estimation step that can be readily applied also to existing VAEs, improving their sample quality. We show, in a rigorous empirical study, that the proposed regularized deterministic autoencoders are able to generate samples that are comparable to, or better than, those of VAEs and more powerful alternatives when applied to images as well as to structured data such as molecules. [1]

## 1 INTRODUCTION

Generative models lie at the core of machine learning. By capturing the mechanisms behind the data generation process, one can reason about data probabilistically, access and traverse the low-dimensional manifold the data is assumed to live on, and ultimately *generate new data*. It is therefore not surprising that generative models have gained momentum in applications such as computer vision (Sohn et al., 2015; Brock et al., 2019), NLP (Bowman et al., 2016; Severyn et al., 2017), and chemistry (Kusner et al., 2017; Jin et al., 2018; Gómez-Bombarelli et al., 2018).

Variational Autoencoders (VAEs) (Kingma & Welling, 2014; Rezende et al., 2014) cast learning representations for high-dimensional distributions as a variational inference problem. Learning a VAE amounts to the optimization of an objective balancing the quality of samples that are autoencoded through a stochastic encoder–decoder pair while encouraging the latent space to follow a fixed prior distribution. Since their introduction, VAEs have become one of the frameworks of choice among the different generative models. VAEs promise theoretically well-founded and more stable training than Generative Adversarial Networks (GANs) (Goodfellow et al., 2014) and more efficient sampling mechanisms than autoregressive models (Larochelle & Murray, 2011; Germain et al., 2015).

However, the VAE framework is still far from delivering the promised generative mechanism, as there are several practical and theoretical challenges yet to be solved. A major weakness of VAEs is

---

*Equal contribution.

[1]An implementation is available at: `https://github.com/ParthaEth/Regularized_autoencoders-RAE-`

the tendency to strike an unsatisfying compromise between sample quality and reconstruction quality. In practice, this has been attributed to overly simplistic prior distributions (Tomczak & Welling, 2018; Dai & Wipf, 2019) or alternatively, to the inherent over-regularization induced by the KL divergence term in the VAE objective (Tolstikhin et al., 2017). Most importantly, the VAE objective itself poses several challenges as it admits trivial solutions that decouple the latent space from the input (Chen et al., 2017; Zhao et al., 2017), leading to the posterior collapse phenomenon in conjunction with powerful decoders (van den Oord et al., 2017). Furthermore, due to its variational formulation, training a VAE requires approximating expectations through sampling at the cost of increased variance in gradients (Burda et al., 2015; Tucker et al., 2017), making initialization, validation, and annealing of hyperparameters essential in practice (Bowman et al., 2016; Higgins et al., 2017; Bauer & Mnih, 2019). Lastly, even after a satisfactory convergence of the objective, the learned aggregated posterior distribution rarely matches the assumed latent prior in practice (Kingma et al., 2016; Bauer & Mnih, 2019; Dai & Wipf, 2019), ultimately hurting the quality of generated samples. All in all, much of the attention around VAEs is still directed towards "fixing" the aforementioned drawbacks associated with them.

In this work, we take a different route: we question whether the variational framework adopted by VAEs is necessary for generative modeling and, in particular, to obtain a smooth latent space. We propose to adopt a simpler, deterministic version of VAEs that scales better, is simpler to optimize, and, most importantly, still produces a meaningful latent space and equivalently good or better samples than VAEs or stronger alternatives, e.g., Wasserstein Autoencoders (WAEs) (Tolstikhin et al., 2017). We do so by observing that, under commonly used distributional assumptions, training a stochastic encoder–decoder pair in VAEs does not differ from training a deterministic architecture where noise is added to the decoder's input. We investigate how to substitute this noise injection mechanism with other regularization schemes in the proposed deterministic *Regularized Autoencoders* (RAEs), and we thoroughly analyze how this affects performance. Finally, we equip RAEs with a generative mechanism via a simple ex-post density estimation step on the learned latent space.

In summary, our contributions are as follows: i) we introduce the RAE framework for generative modeling as a drop-in replacement for many common VAE architectures; ii) we propose an ex-post density estimation scheme which greatly improves sample quality for VAEs, WAEs and RAEs without the need to retrain the models; iii) we conduct a rigorous empirical evaluation to compare RAEs with VAEs and several baselines on standard image datasets and on more challenging structured domains such as molecule generation (Kusner et al., 2017; Gómez-Bombarelli et al., 2018).

## 2 VARIATIONAL AUTOENCODERS

For a general discussion, we consider a collection of high-dimensional i.i.d. samples $\mathcal{X} = \{\mathbf{x}_i\}_{i=1}^N$ drawn from the true data distribution $p_{\mathsf{data}}(\mathbf{x})$ over a random variable $\mathbf{X}$ taking values in the input space. The aim of generative modeling is to learn from $\mathcal{X}$ a mechanism to draw new samples $\mathbf{x}_{\mathsf{new}} \sim p_{\mathsf{data}}$. Variational Autoencoders provide a powerful latent variable framework to infer such a mechanism. The generative process of the VAE is defined as

$$\mathbf{z}_{\mathsf{new}} \sim p(\mathbf{Z}), \qquad \mathbf{x}_{\mathsf{new}} \sim p_\theta(\mathbf{X} \,|\, \mathbf{Z} = \mathbf{z}_{\mathsf{new}}) \tag{1}$$

where $p(\mathbf{Z})$ is a fixed prior distribution over a low-dimensional latent space $\mathbf{Z}$. A stochastic decoder

$$D_\theta(\mathbf{z}) = \mathbf{x} \sim p_\theta(\mathbf{x} \,|\, \mathbf{z}) = p(\mathbf{X} \,|\, g_\theta(\mathbf{z})) \tag{2}$$

links the latent space to the input space through the *likelihood* distribution $p_\theta$, where $g_\theta$ is an expressive non-linear function parameterized by $\theta$.[2] As a result, a VAE estimates $p_{\mathsf{data}}(\mathbf{x})$ as the infinite mixture model $p_\theta(\mathbf{x}) = \int p_\theta(\mathbf{x} \,|\, \mathbf{z}) p(\mathbf{z}) d\mathbf{z}$. At the same time, the input space is mapped to the latent space via a stochastic encoder

$$E_\phi(\mathbf{x}) = \mathbf{z} \sim q_\phi(\mathbf{z} \,|\, \mathbf{x}) = q(\mathbf{Z} \,|\, f_\phi(\mathbf{x})) \tag{3}$$

where $q_\phi(\mathbf{z} \,|\, \mathbf{x})$ is the *posterior* distribution given by a second function $f_\phi$ parameterized by $\phi$. Computing the marginal log-likelihood $\log p_\theta(\mathbf{x})$ is generally intractable. One therefore follows a variational approach, maximizing the evidence lower bound (ELBO) for a sample $\mathbf{x}$:

$$\log p_\theta(\mathbf{x}) \geq \mathsf{ELBO}(\phi, \theta, \mathbf{x}) = \mathbb{E}_{\mathbf{z} \sim q_\phi(\mathbf{z} \,|\, \mathbf{x})} \log p_\theta(\mathbf{x} \,|\, \mathbf{z}) - \mathbb{KL}(q_\phi(\mathbf{z} \,|\, \mathbf{x}) || p(\mathbf{z})) \tag{4}$$

---

[2]With slight abuse of notation, we use lowercase letters for both random variables and their realizations, e.g., $p_\theta(\mathbf{x} \,|\, \mathbf{z})$ instead of $p(\mathbf{X} \,|\, \mathbf{Z} = \mathbf{z})$, when it is clear to discriminate between the two.

Maximizing Eq. 4 over data $\mathcal{X}$ w.r.t. model parameters $\phi$, $\theta$ corresponds to minimizing the loss

$$\arg\min_{\phi,\theta} \mathbb{E}_{\mathbf{x}\sim p_{\text{data}}} \mathcal{L}_{\text{ELBO}} = \mathbb{E}_{\mathbf{x}\sim p_{\text{data}}} \mathcal{L}_{\text{REC}} + \mathcal{L}_{\text{KL}} \tag{5}$$

where $\mathcal{L}_{\text{REC}}$ and $\mathcal{L}_{\text{KL}}$ are defined for a sample $\mathbf{x}$ as follows:

$$\mathcal{L}_{\text{REC}} = -\mathbb{E}_{\mathbf{z}\sim q_\phi(\mathbf{z}\,|\,\mathbf{x})} \log p_\theta(\mathbf{x}\,|\,\mathbf{z}) \qquad \mathcal{L}_{\text{KL}} = \mathbb{KL}(q_\phi(\mathbf{z}\,|\,\mathbf{x})||p(\mathbf{z})) \tag{6}$$

Intuitively, the reconstruction loss $\mathcal{L}_{\text{REC}}$ takes into account the quality of autoencoded samples $\mathbf{x}$ through $D_\theta(E_\phi(\mathbf{x}))$, while the KL-divergence term $\mathcal{L}_{\text{KL}}$ encourages $q_\phi(\mathbf{z}\,|\,\mathbf{x})$ to match the prior $p(\mathbf{z})$ for each $\mathbf{z}$ which acts as a regularizer during training (Hoffman & Johnson, 2016).

## 2.1 PRACTICE AND SHORTCOMINGS OF VAEs

To fit a VAE to data through Eq. 5 one has to specify the parametric forms for $p(\mathbf{z})$, $q_\phi(\mathbf{z}\,|\,\mathbf{x})$, $p_\theta(\mathbf{x}\,|\,\mathbf{z})$, and hence the deterministic mappings $f_\phi$ and $g_\theta$. In practice, the choice for the above distributions is guided by trading off computational complexity with model expressiveness. In the most commonly adopted formulation of the VAE, $q_\phi(\mathbf{z}\,|\,\mathbf{x})$ and $p_\theta(\mathbf{x}\,|\,\mathbf{z})$ are assumed to be Gaussian:

$$E_\phi(\mathbf{x}) \sim \mathcal{N}(\mathbf{Z}|\boldsymbol{\mu}_\phi(\mathbf{x}), \text{diag}(\boldsymbol{\sigma}_\phi(\mathbf{x}))) \qquad D_\theta(E_\phi(\mathbf{x})) \sim \mathcal{N}(\mathbf{X}|\boldsymbol{\mu}_\theta(\mathbf{z}), \text{diag}(\boldsymbol{\sigma}_\theta(\mathbf{z}))) \tag{7}$$

with means $\mu_\phi, \mu_\theta$ and covariance parameters $\sigma_\phi, \sigma_\theta$ given by $f_\phi$ and $g_\theta$. In practice, the covariance of the decoder is set to the identity matrix for all $\mathbf{z}$, i.e., $\sigma_\theta(\mathbf{z}) = 1$ (Dai & Wipf, 2019). The expectation of $\mathcal{L}_{\text{REC}}$ in Eq. 6 must be approximated via $k$ Monte Carlo point estimates. It is expected that the quality of the Monte Carlo estimate, and hence convergence during learning and sample quality increases for larger $k$ (Burda et al., 2015). However, only a 1-sample approximation is generally carried out (Kingma & Welling, 2014) since memory and time requirements are prohibitive for large $k$. With the 1-sample approximation, $\mathcal{L}_{\text{REC}}$ can be computed as the mean squared error between input samples and their mean reconstructions $\mu_\theta$ by a decoder that is deterministic in practice:

$$\mathcal{L}_{\text{REC}} = ||\mathbf{x} - \boldsymbol{\mu}_\theta(E_\phi(\mathbf{x}))||_2^2 \tag{8}$$

Gradients w.r.t. the encoder parameters $\phi$ are computed through the expectation of $\mathcal{L}_{\text{REC}}$ in Eq. 6 via the reparametrization trick (Kingma & Welling, 2014) where the stochasticity of $E_\phi$ is relegated to an auxiliary random variable $\epsilon$ which does not depend on $\phi$:

$$E_\phi(\mathbf{x}) = \boldsymbol{\mu}_\phi(\mathbf{x}) + \boldsymbol{\sigma}_\phi(\mathbf{x}) \odot \boldsymbol{\epsilon}, \qquad \boldsymbol{\epsilon} \sim \mathcal{N}(\mathbf{0}, \mathbf{I}) \tag{9}$$

where $\odot$ denotes the Hadamard product. An additional simplifying assumption involves fixing the prior $p(\mathbf{z})$ to be a $d$-dimensional isotropic Gaussian $\mathcal{N}(\mathbf{Z}\,|\,\mathbf{0}, \mathbf{I})$. For this choice, the KL-divergence for a sample $\mathbf{x}$ is given in closed form: $2\mathcal{L}_{\text{KL}} = ||\boldsymbol{\mu}_\phi(\mathbf{x})||_2^2 + d + \sum_i^d \boldsymbol{\sigma}_\phi(\mathbf{x})_i - \log \boldsymbol{\sigma}_\phi(\mathbf{x})_i$.

While the above assumptions make VAEs easy to implement, the stochasticity in the encoder and decoder are still problematic in practice (Makhzani et al., 2016; Tolstikhin et al., 2017; Dai & Wipf, 2019). In particular, one has to carefully balance the trade-off between the $\mathcal{L}_{\text{KL}}$ term and $\mathcal{L}_{\text{REC}}$ during optimization (Dai & Wipf, 2019; Bauer & Mnih, 2019). A too-large weight on the $\mathcal{L}_{\text{KL}}$ term can dominate $\mathcal{L}_{\text{ELBO}}$, having the effect of *over-regularization*. As this would smooth the latent space, it can directly affect sample quality in a negative way. Heuristics to avoid this include manually fine-tuning or gradually annealing the importance of $\mathcal{L}_{\text{KL}}$ during training (Bowman et al., 2016; Bauer & Mnih, 2019). We also observe this trade-off in a practical experiment in Appendix A.

Even after employing the full array of approximations and "tricks" to reach convergence of Eq. 5 for a satisfactory set of parameters, there is no guarantee that the learned latent space is distributed according to the assumed prior distribution. In other words, the aggregated posterior distribution $q_\phi(\mathbf{z}) = \mathbb{E}_{\mathbf{x}\sim p_{\text{data}}} q(\mathbf{z}|\mathbf{x})$ has been shown not to conform well to $p(\mathbf{z})$ after training (Tolstikhin et al., 2017; Bauer & Mnih, 2019; Dai & Wipf, 2019). This critical issue severely hinders the generative mechanism of VAEs (cf. Eq. 1) since latent codes sampled from $p(\mathbf{z})$ (instead of $q(\mathbf{z})$) might lead to regions of the latent space that are previously unseen to $D_\theta$ during training. This results in generating out-of-distribution samples. We refer the reader to Appendix H for a visual demonstration of this phenomenon on the latent space of VAEs. We analyze solutions to this problem in Section 4.

## 2.2 Constant-Variance Encoders

Before introducing our fully-deterministic take on VAEs, it is worth investigating intermediate flavors of VAEs with reduced stochasticity. Analogous to what is commonly done for decoders as discussed in the previous section, one can fix the variance of $q_\phi(\mathbf{z} \,|\, \mathbf{x})$ to be constant for all $\mathbf{x}$. This simplifies the computation of $E_\phi$ from Eq. 9 to

$$E_\phi^{\mathsf{CV}}(\mathbf{x}) = \boldsymbol{\mu}_\phi(\mathbf{x}) + \boldsymbol{\epsilon}, \qquad \boldsymbol{\epsilon} \sim \mathcal{N}(\mathbf{0}, \sigma\mathbf{I}) \tag{10}$$

where $\sigma$ is a fixed scalar. Then, the KL loss term in a Gaussian VAE simplifies (up to a constant) to $\mathcal{L}_{\mathsf{KL}}^{\mathsf{CV}} = ||\boldsymbol{\mu}_\phi(\mathbf{x})||_2^2$. We name this variant Constant-Variance VAEs (CV-VAEs). While CV-VAEs have been adopted in some applications such as variational image compression (Ballé et al., 2017) and adversarial robustness (Ghosh et al., 2019), to the best of our knowledge, there is no systematic study of them in the literature. We will fill this gap in our experiments in Section 6. Lastly, note that now $\sigma$ in Eq.10 is not learned along the encoder as in Eq. 9. Nevertheless, it can still be fitted as an hyperparameter, e.g., by cross-validation, to maximise the model likelihood. This highlights the possibility to estimate a better parametric form for the latent space distribution after training, or in a outer-loop including training. We address this provide a more complex and flexible solution to deal with the prior structure over $\mathbf{Z}$ via ex-post density estimation in Section 4.

## 3 Deterministic Regularized Autoencoders

Autoencoding in VAEs is defined in a probabilistic fashion: $E_\phi$ and $D_\theta$ map data points not to a single point, but rather to parameterized distributions (cf. Eq. 7). However, common implementations of VAEs as discussed in Section 2 admit a simpler, deterministic view for this probabilistic mechanism. A glance at the autoencoding mechanism of the VAE is revealing.

The encoder deterministically maps a data point $\mathbf{x}$ to mean $\mu_\phi(\mathbf{x})$ and variance $\sigma_\phi(\mathbf{x})$ in the latent space. The input to $D_\theta$ is then simply the mean $\mu_\phi(\mathbf{x})$ *augmented with Gaussian noise* scaled by $\sigma_\phi(\mathbf{x})$ via the reparametrization trick (cf. Eq. 9). In the CV-VAE, this relationship is even more obvious, as the magnitude of the noise is fixed for all data points (cf. Eq. 10). In this light, a VAE can be seen as a *deterministic* autoencoder where (Gaussian) noise is added to the decoder's input.

We argue that this noise injection mechanism is a key factor in having a regularized decoder. Using random noise injection to regularize neural networks is a well-known technique that dates back several decades (Sietsma & Dow, 1991; An, 1996). It implicitly helps to smooth the function learned by the network at the price of increased variance in the gradients during training. In turn, decoder regularization is a key component in generalization for VAEs, as it improves random sample quality and achieves a smoother latent space. Indeed, from a generative perspective, regularization is motivated by the goal to learn a smooth latent space where similar data points $\mathbf{x}$ are mapped to similar latent codes $\mathbf{z}$, and small variations in $\mathbf{Z}$ lead to reconstructions by $D_\theta$ that vary only slightly.

We propose to *substitute noise injection with an explicit regularization scheme for the decoder*. This entails the substitution of the variational framework in VAEs, which enforces regularization on the encoder posterior through $\mathcal{L}_{\mathsf{KL}}$, with a deterministic framework that applies other flavors of decoder regularization. By removing noise injection from a CV-VAE, we are effectively left with a deterministic autoencoder (AE). Coupled with explicit regularization for the decoder, we obtain a *Regularized Autoencoder* (RAE). Training a RAE thus involves minimizing the simplified loss

$$\mathcal{L}_{\mathsf{RAE}} = \mathcal{L}_{\mathsf{REC}} + \beta\mathcal{L}_{\mathbf{Z}}^{\mathsf{RAE}} + \lambda\mathcal{L}_{\mathsf{REG}} \tag{11}$$

where $\mathcal{L}_{\mathsf{REG}}$ represents the explicit regularizer for $D_\theta$ (discussed in Section 3.1) and $\mathcal{L}_{\mathbf{Z}}^{\mathsf{RAE}} = {}^1\!/_2||\mathbf{z}||_2^2$ (resulting from simplifying $\mathcal{L}_{\mathsf{KL}}^{\mathsf{CV}}$) is equivalent to constraining the size of the learned latent space, which is still needed to prevent unbounded optimization. Finally, $\beta$ and $\lambda$ are two hyper parameters that balance the different loss terms.

Note that for RAEs, no Monte Carlo approximation is required to compute $\mathcal{L}_{\mathsf{REC}}$. This relieves the need for more samples from $q_\phi(\mathbf{z} \,|\, \mathbf{x})$ to achieve better image quality (cf. Appendix A). Moreover, by abandoning the variational framework and the $\mathcal{L}_{\mathsf{KL}}$ term, there is no need in RAEs for a fixed prior distribution over $\mathbf{Z}$. Doing so however loses a clear generative mechanism for RAEs to sample from $\mathbf{Z}$. We propose a method to regain random sampling ability in Section 4 by performing density estimation on $\mathbf{Z}$ ex-post, a step that is otherwise still needed for VAEs to alleviate the posterior mismatch issue.

### 3.1 REGULARIZATION SCHEMES FOR RAES

Among possible choices for $\mathcal{L}_{\mathsf{REG}}$, a first obvious candidate is Tikhonov regularization (Tikhonov & Arsenin, 1977) since is known to be related to the addition of low-magnitude input noise (Bishop, 2006). Training a RAE within this framework thus amounts to adopting $\mathcal{L}_{\mathsf{REG}} = \mathcal{L}_{\mathsf{L}_2} = ||\theta||_2^2$ which effectively applies weight decay on the decoder parameters $\theta$.

Another option comes from the recent GAN literature where regularization is a hot topic (Kurach et al., 2018) and where injecting noise to the input of the adversarial discriminator has led to improved performance in a technique called *instance noise* (Sønderby et al., 2017). To enforce Lipschitz continuity on adversarial discriminators, weight clipping has been proposed (Arjovsky et al., 2017), which is however known to significantly slow down training. More successfully, a *gradient penalty* on the discriminator can be used similar to Gulrajani et al. (2017); Mescheder et al. (2018), yielding the objective $\mathcal{L}_{\mathsf{REG}} = \mathcal{L}_{\mathsf{GP}} = ||\nabla D_\theta(E_\phi(\mathbf{x}))||_2^2$ which bounds the gradient norm of the decoder w.r.t. its input.

Additionally, spectral normalization (SN) has been successfully proposed as an alternative way to bound the Lipschitz norm of an adversarial discriminator (Miyato et al., 2018). SN normalizes each weight matrix $\theta_\ell$ in the decoder by an estimate of its largest singular value: $\theta_\ell^{\mathsf{SN}} = \theta_\ell/\mathsf{s}(\theta_\ell)$ where $\mathsf{s}(\theta_\ell)$ is the current estimate obtained through the power method.

In light of the recent successes of deep networks *without* explicit regularization (Zagoruyko & Komodakis, 2016; Zhang et al., 2017), it is intriguing to question the need for explicit regularization of the decoder in order to obtain a meaningful latent space. The assumption here is that techniques such as dropout (Srivastava et al., 2014), batch normalization (Ioffe & Szegedy, 2015), adding noise during training (An, 1996) implicitly regularize the networks enough. Therefore, as a natural baseline to the $\mathcal{L}_{\mathsf{RAE}}$ objectives introduced above, we also consider the RAE framework without $\mathcal{L}_{\mathsf{REG}}$ and $\mathcal{L}_{\mathbf{Z}}^{\mathsf{RAE}}$, i.e., a standard deterministic autoencoder optimizing $\mathcal{L}_{\mathsf{REC}}$ only.

To complete our "autopsy" of the VAE loss, we additionally investigate deterministic autoencoders with decoder regularization, but without the $\mathcal{L}_{\mathbf{Z}}^{\mathsf{RAE}}$ term, as well as possible combinations of different regularizers in our RAE framework (cf. Table 3 in Appendix I).

Lastly, it is worth questioning if it is possible to formally derive our RAE framework from first principles. We answer this affirmatively, and show how to augment the ELBO optimization problem of a VAE with an explicit constraint, while not fixing a parametric form for $q_\phi(\mathbf{z} \,|\, \mathbf{x})$. This indeed leads to a special case of the RAE loss in Eq. 11. Specifically, we derive a regularizer like $\mathcal{L}_{\mathsf{GP}}$ for a deterministic version of the CV-VAE. Note that this derivation legitimates bounding the decoder's gradients and as such it justifies the spectral norm regularizer as well since the latter enforces the decoder's Lipschitzness. We accommodate the full derivation in Appendix B.

## 4   EX-POST DENSITY ESTIMATION

By removing stochasticity and ultimately, the KL divergence term $\mathcal{L}_{\mathsf{KL}}$ from RAEs, we have simplified the original VAE objective at the cost of detaching the encoder from the prior $p(\mathbf{z})$ over the latent space. This implies that i) we cannot ensure that the latent space $\mathbf{Z}$ is distributed according to a simple distribution (e.g., isotropic Gaussian) anymore and consequently, ii) we lose the simple mechanism provided by $p(\mathbf{z})$ to sample from $\mathbf{Z}$ as in Eq. 1.

As discussed in Section 2.1, issue i) is compromising the VAE framework in any case, as reported in several works (Hoffman & Johnson, 2016; Rosca et al., 2018; Dai & Wipf, 2019). To fix this, some works extend the VAE objective by encouraging the aggregated posterior to match $p(\mathbf{z})$ (Tolstikhin et al., 2017) or by utilizing more complex priors (Kingma et al., 2016; Tomczak & Welling, 2018; Bauer & Mnih, 2019).

To overcome both i) and ii), we instead propose to employ *ex-post density estimation* over $\mathbf{Z}$. We fit a density estimator denoted as $q_\delta(\mathbf{z})$ to $\{\mathbf{z} = E_\phi(\mathbf{x})|\mathbf{x} \in \mathcal{X}\}$. This simple approach not only fits our RAE framework well, but it can also be readily adopted for any VAE or variants thereof such as the WAE as a practical remedy to the aggregated posterior mismatch without adding any computational overhead to the costly training phase.

The choice of $q_\delta(\mathbf{z})$ needs to trade-off *expressiveness* – to provide a good fit of an arbitrary space for $\mathbf{Z}$ – with *simplicity*, to improve generalization. For example, placing a Dirac distribution on each latent point $\mathbf{z}$ would allow the decoder to output only training sample reconstructions which have a high quality, but do not generalize. Striving for simplicity, we employ and compare a full covariance multivariate Gaussian with a 10-component Gaussian mixture model (GMM) in our experiments.

## 5 RELATED WORKS

Many works have focused on diagnosing the VAE framework, the terms in its objective (Hoffman & Johnson, 2016; Zhao et al., 2017; Alemi et al., 2018), and ultimately augmenting it to solve optimization issues (Rezende & Viola, 2018; Dai & Wipf, 2019). With RAE, we argue that a simpler deterministic framework can be competitive for generative modeling.

Deterministic denoising (Vincent et al., 2008) and contractive autoencoders (CAEs) (Rifai et al., 2011) have received attention in the past for their ability to capture a smooth data manifold. Heuristic attempts to equip them with a generative mechanism include MCMC schemes (Rifai et al., 2012; Bengio et al., 2013). However, they are hard to diagnose for convergence, require a considerable effort in tuning (Cowles & Carlin, 1996), and have not scaled beyond MNIST, leading to them being superseded by VAEs. While computing the Jacobian for CAEs (Rifai et al., 2011) is close in spirit to $\mathcal{L}_{\text{GP}}$ for RAEs, the latter is much more computationally efficient.

Approaches to cope with the aggregated posterior mismatch involve fixing a more expressive form for $p(\mathbf{z})$ (Kingma et al., 2016; Bauer & Mnih, 2019) therefore altering the VAE objective and requiring considerable additional computational efforts. Estimating the latent space of a VAE with a second VAE (Dai & Wipf, 2019) reintroduces many of the optimization shortcomings discussed for VAEs and is much more expensive in practice compared to fitting a simple $q_\delta(\mathbf{z})$ after training.

Adversarial Autoencoders (AAE) (Makhzani et al., 2016) add a discriminator to a deterministic encoder–decoder pair, leading to sharper samples at the expense of higher computational overhead and the introduction of instabilities caused by the adversarial nature of the training process.

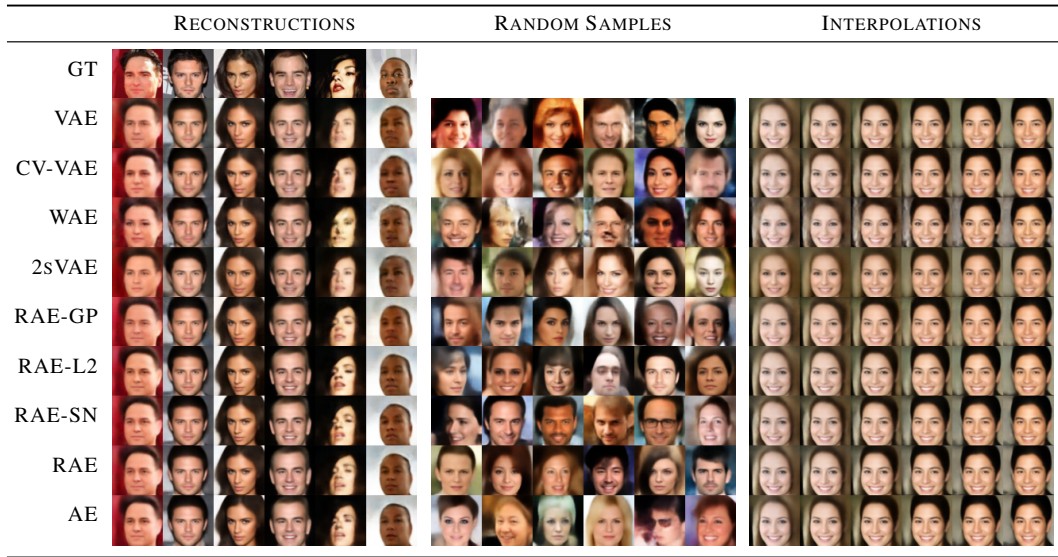

Figure 1: Qualitative evaluation of sample quality for VAEs, WAEs, 2sVAEs, and RAEs on CelebA. RAE provides slightly sharper samples and reconstructions while interpolating smoothly in the latent space. Corresponding qualitative overviews for MNIST and CIFAR-10 are provided in Appendix F.

Wasserstein Autoencoders (WAE) (Tolstikhin et al., 2017) have been introduced as a generalization of AAEs by casting autoencoding as an optimal transport (OT) problem. Both stochastic and deterministic models can be trained by minimizing a relaxed OT cost function employing either an adversarial loss term or the maximum mean discrepancy score between $p(\mathbf{z})$ and $q_\phi(\mathbf{z})$ as a reg-

ularizer in place of $\mathcal{L}_{\mathsf{KL}}$. Within the RAE framework, we look at this problem from a different perspective: instead of explicitly imposing a simple structure on $\mathbf{Z}$ that might impair the ability to fit high-dimensional data during training, we propose to model the latent space by an ex-post density estimation step.

The most successful VAE architectures for images and audio so far are variations of the VQ-VAE (van den Oord et al., 2017; Razavi et al., 2019). Despite the name, VQ-VAEs are neither stochastic, nor variational, but they are deterministic autoencoders. VQ-VAEs are similar to RAEs in that they adopt ex-post density estimation. However, VQ-VAEs necessitates complex discrete autoregressive density estimators and a training loss that is non-differentiable due to quantizing $\mathbf{Z}$.

Lastly, RAEs share some similarities with GLO (**?**). However, differently from RAEs, GLO can be interpreted as a deterministic AE without and encoder, and when the latent space is built "on-demand" by optimization. On the other hand, RAEs augment deterministic decoders as in GANs with deterministic encoders.

## 6  EXPERIMENTS

Our experiments are designed to answer the following questions: **Q1:** Are sample quality and latent space structure in RAEs comparable to VAEs? **Q2:** How do different regularizations impact RAE performance? **Q3:** What is the effect of ex-post density estimation on VAEs and its variants?

| | MNIST | | | | CIFAR | | | | CELEBA | | | |
|---|---|---|---|---|---|---|---|---|---|---|---|---|
| | | SAMPLES | | | | SAMPLES | | | | SAMPLES | | |
| | REC. | $\mathcal{N}$ | GMM | Interp. | REC. | $\mathcal{N}$ | GMM | Interp. | REC. | $\mathcal{N}$ | GMM | Interp. |
| VAE | 18.26 | 19.21 | 17.66 | 18.21 | 57.94 | 106.37 | 103.78 | 88.62 | 39.12 | 48.12 | 45.52 | 44.49 |
| CV-VAE | 15.15 | 33.79 | 17.87 | 25.12 | 37.74 | 94.75 | 86.64 | 69.71 | 40.41 | 48.87 | 49.30 | 44.96 |
| WAE | **10.03** | 20.42 | 9.39 | **14.34** | 35.97 | 117.44 | 93.53 | 76.89 | **34.81** | 53.67 | 42.73 | 40.93 |
| 2sVAE | 20.31 | **18.81** | – | 18.35 | 62.54 | 109.77 | – | 89.06 | 42.04 | 49.70 | – | 47.54 |
| RAE-GP | 14.04 | 22.21 | 11.54 | 15.32 | 32.17 | 83.05 | 76.33 | 64.08 | 39.71 | 116.30 | 45.63 | 47.00 |
| RAE-L2 | 10.53 | 22.22 | **8.69** | 14.54 | 32.24 | **80.80** | **74.16** | 62.54 | 43.52 | 51.13 | 47.97 | 45.98 |
| RAE-SN | 15.65 | 19.67 | 11.74 | 15.15 | **27.61** | 84.25 | 75.30 | 63.62 | 36.01 | **44.74** | **40.95** | **39.53** |
| RAE | 11.67 | 23.92 | 9.81 | 14.67 | 29.05 | 83.87 | 76.28 | 63.27 | 40.18 | 48.20 | 44.68 | 43.67 |
| AE | 12.95 | 58.73 | 10.66 | 17.12 | 30.52 | 84.74 | 76.47 | **61.57** | 40.79 | 127.85 | 45.10 | 50.94 |
| AE-L2 | 11.19 | 315.15 | 9.36 | 17.15 | 34.35 | 247.48 | 75.40 | 61.09 | 44.72 | 346.29 | 48.42 | 56.16 |

Table 1: Evaluation of all models by FID (lower is better, best models in bold). We evaluate each model by REC.: test sample reconstruction; $\mathcal{N}$: random samples generated according to the prior distribution $p(\mathbf{z})$ (isotropic Gaussian for VAE / WAE, another VAE for 2SVAE) or by fitting a Gaussian to $q_\delta(\mathbf{z})$ (for the remaining models); GMM: random samples generated by fitting a mixture of 10 Gaussians in the latent space; Interp.: mid-point interpolation between random pairs of test reconstructions. The RAE models are competitive with or outperform previous models throughout the evaluation. Interestingly, interpolations do not suffer from the lack of explicit priors on the latent space in our models.

### 6.1  RAES FOR IMAGE MODELING

We evaluate all regularization schemes from Section 3.1: RAE-GP, RAE-L2, and RAE-SN. For a thorough ablation study, we also consider only adding the latent code regularizer $\mathcal{L}_{\mathbf{Z}}^{\mathsf{RAE}}$ to $\mathcal{L}_{\mathsf{REC}}$ (RAE), and an autoencoder without any explicit regularization (AE). We check the effect of applying one regularization scheme while not including the $\mathcal{L}_{\mathbf{Z}}^{\mathsf{RAE}}$ term in the AE-L2 model.

As baselines, we employ the regular VAE, constant-variance VAE (CV-VAE), Wasserstein Autoencoder (WAE) with the MMD loss as a state-of-the-art method, and the recent 2-stage VAE (2sVAE) (Dai & Wipf, 2019) which performs a form of ex-post density estimation via another VAE. For a fair comparison, we use the same network architecture for all models. Further details about the architecture and training are given in Appendix C.

We measure the following quantities: held-out sample reconstruction quality, random sample quality, and interpolation quality. While reconstructions give us a lower bound on the best quality

achievable by the generative model, random sample quality indicates how well the model generalizes. Finally, interpolation quality sheds light on the structure of the learned latent space. The evaluation of generative models is a nontrivial research question (Theis et al., 2016; Sajjadi et al., 2017; Lucic et al., 2018). We report here the ubiquitous Fréchet Inception Distance (FID) (Heusel et al., 2017) and we provide precision and recall scores (PRD) (Sajjadi et al., 2018) in Appendix E.

Table 1 summarizes our main results. All of the proposed RAE variants are competitive with the VAE, WAE and 2sVAE w.r.t. generated image quality in all settings. Sampling RAEs achieve the best FIDs across all datasets when a modest 10-component GMM is employed for ex-post density estimation. Furthermore, even when $\mathcal{N}$ is considered as $q_\delta(\mathbf{z})$, RAEs rank first with the exception of MNIST, where it competes for the second position with a VAE. Our best RAE FIDs are lower than the best results reported for VAEs in the large scale comparison of (Lucic et al., 2018), challenging even the best scores reported for GANs. While we are employing a slightly different architecture than theirs, our models underwent only modest finetuning instead of an extensive hyperparameter search. A comparison of the different regularization schemes for RAEs (Q2) yields no clear winner across all settings as all perform equally well. Striving for a simpler implementation, one may prefer RAE-L2 over the GP and SN variants.

For completeness, we investigate applying multiple regularization schemes to our RAE models. We report the results of all possible combinations in Table 3, Appendix I. There, no significant boost of performance can be spotted when comparing to singly regularized RAEs.

Surprisingly, the implicitly regularized RAE and AE models are shown to be able to score impressive FIDs when $q_\delta(\mathbf{z})$ is fit through GMMs. FIDs for AEs decrease from 58.73 to 10.66 on MNIST and from 127.85 to 45.10 on CelebA – a value close to the state of the art. This is a remarkable result that follows a long series of recent confirmations that neural networks are surprisingly smooth by design (Neyshabur et al., 2017). It is also surprising that the lack of an explicitly fixed structure on the latent space of the RAE does not impede interpolation quality. This is further confirmed by the qualitative evaluation on CelebA as reported in Fig. 1 and for the other datasets in Appendix F, where RAE interpolated samples seem sharper than competitors and transitions smoother.

Our results further confirm and quantify the effect of the aggregated posterior mismatch. In Table 1, ex-post density estimation consistently improves sample quality across all settings and models. A 10-component GMM halves FID scores from $\sim$20 to $\sim$10 for WAE and RAE models on MNIST and from 116 to 46 on CelebA. This is especially striking since this additional step is much cheaper and simpler than training a second-stage VAE as in 2sVAE (Q3). In summary, the results strongly support the conjecture that the simple deterministic RAE framework can challenge VAEs and stronger alternatives (Q1).

## 6.2 GRAMMARRAE: MODELING STRUCTURED INPUTS

We now evaluate RAEs for generating complex structured objects such as molecules and arithmetic expressions. We do this with a twofold aim: i) to investigate the latent space learned by RAE for more challenging input spaces that abide to some structural constraints, and ii) to quantify the gain of replacing the VAE in a state-of-the-art generative model with a RAE.

To this end, we adopt the exact architectures and experimental settings of the GrammarVAE (GVAE) (Kusner et al., 2017), which has been shown to outperform other generative alternatives such as the CharacterVAE (CVAE) (Gómez-Bombarelli et al., 2018). As in Kusner et al. (2017), we are interested in traversing the latent space learned by our models to generate samples (molecules or expressions) that best fit some downstream metric. This is done by Bayesian optimization (BO) by considering the $\log(1 + \text{MSE})$ (lower is better) for the generated expressions w.r.t. some ground truth points, and the water-octanol partition coefficient ($\log P$) (Pyzer-Knapp et al., 2015) (higher is better) in the case of molecules. A well-behaved latent space will not only generate molecules or expressions with better scores during the BO step, but it will also contain syntactically valid ones, i.e., , samples abide to a grammar of rules describing the problem.

Figure 2 summarizes our results over 5 trials of BO. Our GRAEs (Grammar RAE) achieve better average scores than CVAEs and GVAEs in generating expressions and molecules. This is visible also for the three best samples and their scores for all models, with the exception of the first best expression of GVAE. We include in the comparison also the GCVVAE, the equivalent of a CV-VAE

| Problem | Model | % Valid | Avg. score |
|---|---|---|---|
| Expressions | GRAE | **1.00 ± 0.00** | 3.22 ± 0.03 |
| | GCVVAE | 0.99 ± 0.01 | **2.85 ± 0.08** |
| | GVAE | 0.99 ± 0.01 | 3.26 ± 0.20 |
| | CVAE | 0.82 ± 0.07 | 4.74 ± 0.25 |
| Molecules | GRAE | 0.72 ± 0.09 | **-5.62 ± 0.71** |
| | GCVVAE | **0.76 ± 0.06** | -6.40 ± 0.80 |
| | GVAE | 0.28 ± 0.04 | -7.89 ± 1.90 |
| | CVAE | 0.16 ± 0.04 | -25.64 ± 6.35 |

| Model | # | Expression | Score |
|---|---|---|---|
| GRAE | 1 | $\sin(3) + x$ | 0.39 |
| | 2 | $x + 1/\exp(1)$ | **0.39** |
| | 3 | $x + 1 + 2 * \sin(3 + 1 + 2)$ | **0.43** |
| GCVVAE | 1 | $x + \sin(3) * 1$ | 0.39 |
| | 2 | $x/x/3 + x$ | 0.40 |
| | 3 | $\sin(\exp(\exp(1))) + x/2 * 2$ | **0.43** |
| GVAE | 1 | $x/1 + \sin(x) + \sin(x * x)$ | **0.10** |
| | 2 | $1/2 + (x) + \sin(x * x)$ | 0.46 |
| | 3 | $x/2 + \sin(1) + (x/2)$ | 0.52 |
| CVAE | 1 | $x * 1 + \sin(x) + \sin(3 + x)$ | 0.45 |
| | 2 | $x/1 + \sin(1) + \sin(2 * 2)$ | 0.48 |
| | 3 | $1/1 + (x) + \sin(1/2)$ | 0.61 |

| Model | 1st | 2nd | 3rd |
|---|---|---|---|
| GRAE | | | |
| Score | **3.74** | **3.52** | **3.14** |
| GCVVAE | | | |
| Score | 3.22 | 2.83 | 2.63 |
| GVAE | | | |
| Score | 3.13 | 3.10 | 2.37 |
| CVAE | | | |
| Score | 2.75 | 0.82 | 0.63 |

Figure 2: Generating structured objects by GVAE, CVAE and GRAE. (Upper left) Percentage of valid samples and their average mean score (see text, Section 6.2). The three best expressions (lower left) and molecules (upper right) and their scores are reported for all models.

for structured objects, as an additional baseline. We can observe that while the GCVVAE delivers better average scores for the simpler task of generating equations (even though the single three best equations are on par with GRAE), when generating molecules GRAEs deliver samples associated to much higher scores.

More interestingly, while GRAEs are almost equivalent to GVAEs for the easier task of generating expressions, the proportion of syntactically valid molecules for GRAEs greatly improves over GVAEs (from 28% to 72%).

# 7 Conclusion

While the theoretical derivation of the VAE has helped popularize the framework for generative modeling, recent works have started to expose some discrepancies between theory and practice. We have shown that viewing sampling in VAEs as noise injection to enforce smoothness can enable one to distill a deterministic autoencoding framework that is compatible with several regularization techniques to learn a meaningful latent space. We have demonstrated that such an autoencoding framework can generate comparable or better samples than VAEs while getting around the practical drawbacks tied to a stochastic framework. Furthermore, we have shown that our solution of fitting a simple density estimator on the learned latent space consistently improves sample quality both for the proposed RAE framework as well as for VAEs, WAEs, and 2sVAEs which solves the mismatch between the prior and the aggregated posterior in VAEs.

## Acknowledgements

We would like to thank Anant Raj, Matthias Bauer, Paul Rubenstein and Soubhik Sanyal for fruitful discussions.

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

APPENDIX

## A    RECONSTRUCTION AND REGULARIZATION TRADE-OFF

We train a VAE on MNIST while monitoring the test set reconstruction quality by FID. Figure 3 (left) clearly shows the impact of more expensive $k > 1$ Monte Carlo approximations of Eq. 7 on sample quality during training. The commonly used 1-sample approximation is a clear limitation for VAE training.

Figure 3 (right) depicts the inherent trade-off between reconstruction and random sample quality in VAEs. Enforcing structure and smoothness in the latent space of a VAE affects random sample quality in a negative way. In practice, a compromise needs to be made, ultimately leading to subpar performance.

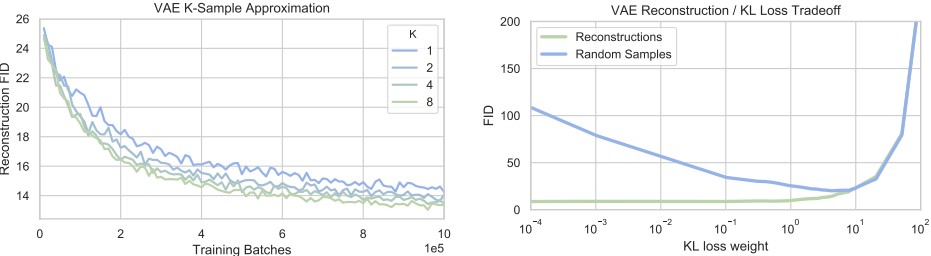

Figure 3: (Left) Test reconstruction quality for a VAE trained on MNIST with different numbers of samples in the latent space as in Eq. 7 measured by FID (lower is better). Larger numbers of Monte-Carlo samples clearly improve training, however, the increased accuracy comes with larger requirements for memory and computation. In practice, the most common choice is therefore $k = 1$. (Right) Reconstruction and random sample quality (FID, y-axis, lower is better) of a VAE on MNIST for different trade-offs between $\mathcal{L}_{\mathsf{REC}}$ and $\mathcal{L}_{\mathsf{KL}}$ (x-axis, see Eq. 5). Higher weights for $\mathcal{L}_{\mathsf{KL}}$ improve random samples but hurt reconstruction. This is especially noticeable towards the optimality point ($\beta \approx 10^1$). This indicates that enforcing structure in the VAE latent space leads to a penalty in quality.

## B    A PROBABILISTIC DERIVATION OF REGULARIZATION

In this section, we propose an alternative view on enforcing smoothness on the output of $D_\theta$ by augmenting the ELBO optimization problem for VAEs with an explicit constraint. While we keep the Gaussianity assumptions over a stochastic $D_\theta$ and $p(\mathbf{z})$ for convenience, we however are not fixing a parametric form for $q_\phi(\mathbf{z} \,|\, \mathbf{x})$ yet. We discuss next how some parametric restrictions over $q_\phi(\mathbf{z} \,|\, \mathbf{x})$ lead to a variation of the RAE framework in Eq. 11, specifically the introduction of $\mathcal{L}_{\mathsf{GP}}$ as a regularizer of a deterministic version of the CV-VAE. To start, we augment Eq. 5 as:

$$\arg\min_{\phi,\theta} \mathbb{E}_{\mathbf{x} \sim p_{\mathrm{data}}(\mathbf{X})} \mathcal{L}_{\mathsf{REC}} + \mathcal{L}_{\mathsf{KL}} \tag{12}$$

$$\text{s.t. } ||D_\theta(\mathbf{z}_1) - D_\theta(\mathbf{z}_2)||_p < \epsilon \quad \forall\, \mathbf{z}_1, \mathbf{z}_2 \sim q_\phi(\mathbf{z} \,|\, \mathbf{x}) \quad \forall \mathbf{x} \sim p_{\mathrm{data}}$$

where $D_\theta(\mathbf{z}) = \boldsymbol{\mu}_\theta(E_\phi(\mathbf{x}))$ and the constraint on the decoder encodes that the output has to vary, in the sense of an $L_p$ norm, only by a small amount $\epsilon$ for any two possible draws from the encoding of $\mathbf{x}$ . Let $D_\theta(\mathbf{z}) : \mathbb{R}^{\dim(\mathbf{z})} \to \mathbb{R}^{\dim(\mathbf{x})}$ be given by a set of $\dim(\mathbf{x})$ given by $\{D_i(\mathbf{z}) : \mathbb{R}^{\dim(\mathbf{z})} \to \mathbb{R}^1\}$. Now we can upper bound the quantity $||D_\theta(\mathbf{z}_1) - D_\theta(\mathbf{z}_2)||_p$ by $\dim(\mathbf{x}) * sup_i\{||D_i(\mathbf{z}_1) - D_i(\mathbf{z}_2)||_p\}$. Using mean value theorem $||D_i(\mathbf{z}_1) - D_i(\mathbf{z}_2)||_p \leq ||\nabla_t D_i((1-t)\mathbf{z}_1 + t\mathbf{z}_2)||_p * ||\mathbf{z}_1 - \mathbf{z}_2||_p$. Hence $sup_i\{||D_i(\mathbf{z}_1) - D_i(\mathbf{z}_2)||_p\} \leq sup_i\{||\nabla_t D_i((1-t)\mathbf{z}_1 + t\mathbf{z}_2)||_p * ||\mathbf{z}_1 - \mathbf{z}_2||_p\}$. Now if we choose the domain of $q_\phi(\mathbf{z} \,|\, \mathbf{x})$ to be isotopic the contribution of $||\mathbf{z}_2 - \mathbf{z}_1||_p$ to the aforementioned quantity becomes a constant factor. Loosely speaking it is the radius of the bounding ball of domain of $q_\phi(\mathbf{z} \,|\, \mathbf{x})$. Hence the above term simplifies to $sup_i\{||\nabla_t D_i((1-t)\mathbf{z}_1 + t\mathbf{z}_2)||_p\}$. Recognizing that here $\mathbf{z}_1$ and $\mathbf{z}_2$ is arbitrary lets us simplify this further to $sup_i\{||\nabla_z D_i(\mathbf{z})||_p\}$

From this form of the smoothness constraint, it is apparent why the choice of a parametric form for $q_\phi(\mathbf{z} \,|\, \mathbf{x})$ can be impactful during training. For a compactly supported isotropic PDF $q_\phi(\mathbf{z}|\mathbf{x})$, the

extension of the support $sup\{||\mathbf{z}_1 - \mathbf{z}_2||_p\}$ would depend on its entropy $\mathbb{H}(q_\phi(\mathbf{z}\,|\,\mathbf{x}))$. through some functional $r$. For instance, a uniform posterior over a hypersphere in $\mathbf{z}$ would ascertain $r(\mathbb{H}(q_\phi(\mathbf{z}\,|\,\mathbf{x}))) \cong e^{\mathbb{H}(q_\phi(\mathbf{z}\,|\,\mathbf{x}))/n}$ where $n$ is the dimensionality of the latent space.

Intuitively, one would look for parametric distributions that do not favor overfitting, e.g., degenerating in Dirac-deltas (minimal entropy and support) along any dimensions. To this end, an isotropic nature of $q_\phi(\mathbf{z}|\mathbf{x})$ would favor such a robustness against decoder over-fitting. We can now rewrite the constraint as

$$r(\mathbb{H}(q_\phi(\mathbf{z}\,|\,\mathbf{x}))) \cdot sup\{||\nabla D_\theta(\mathbf{z})|||_p\} < \epsilon \tag{13}$$

The $\mathcal{L}_{\mathsf{KL}}$ term can be expressed in terms of $\mathbb{H}(q_\phi(\mathbf{z}\,|\,\mathbf{x}))$, by decomposing it as $\mathcal{L}_{\mathsf{KL}} = \mathcal{L}_{\mathsf{CE}} - \mathcal{L}_{\mathsf{H}}$, where $\mathcal{L}_{\mathsf{H}} = \mathbb{H}(q_\phi(\mathbf{z}\,|\,\mathbf{x}))$ and $\mathcal{L}_{\mathsf{CE}} = \mathbb{H}(q_\phi(\mathbf{z}\,|\,\mathbf{x}), p(\mathbf{z}))$ represents a cross-entropy term. Therefore, the constrained problem in Eq. 12 can be written in a Lagrangian formulation by including Eq. 13:

$$\arg\min_{\phi,\theta} \ \mathbb{E}_{\mathbf{x}\sim p_{\mathsf{data}}} \ \mathcal{L}_{\mathsf{REC}} + \mathcal{L}_{\mathsf{CE}} - \mathcal{L}_{\mathsf{H}} + \lambda\mathcal{L}_{\mathsf{LANG}} \tag{14}$$

where $\mathcal{L}_{\mathsf{LANG}} = r(\mathbb{H}(q_\phi(\mathbf{z}\,|\,\mathbf{x}))) * ||\nabla D_\theta(\mathbf{z})||_p$. We argue that a reasonable simplifying assumption for $q_\phi(\mathbf{z}\,|\,\mathbf{x})$ is to fix $\mathbb{H}(q_\phi(\mathbf{z}\,|\,\mathbf{x}))$ to a single constant for all samples $\mathbf{x}$. Intuitively, this can be understood as fixing the variance in $q_\phi(\mathbf{z}\,|\,\mathbf{x})$ as we did for the CV-VAE in Section 2.2. With this simplification, Eq. 14 further reduces to

$$\arg\min_{\phi,\theta} \ \mathbb{E}_{\mathbf{x}\sim p_{\mathsf{data}}(\mathbf{x})} \ \mathcal{L}_{\mathsf{REC}} + \mathcal{L}_{\mathsf{CE}} + \lambda||\nabla D_\theta(\mathbf{z})||_p \tag{15}$$

We can see that $||\nabla D_\theta(\mathbf{z})||_p$ results to be the gradient penalty $\mathcal{L}_{\mathsf{GP}}$ and $\mathcal{L}_{\mathsf{CE}} = ||\mathbf{z}||_2^2$ corresponds to $\mathcal{L}_{\mathsf{KL}}^{\mathsf{RAE}}$, thus recovering our RAE framework as presented in Eq. 11.

## C  NETWORK ARCHITECTURE, TRAINING DETAILS AND EVALUATION

We follow the models adopted by Tolstikhin et al. (2017) with the difference that we consistently apply batch normalization (Ioffe & Szegedy, 2015). The latent space dimension is 16 for MNIST (LeCun et al., 1998), 128 for CIFAR-10 (Krizhevsky & Hinton, 2009) and 64 for CelebA (Liu et al., 2015).

For all experiments, we use the Adam optimizer with a starting learning rate of $10^{-3}$ which is cut in half every time the validation loss plateaus. All models are trained for a maximum of 100 epochs on MNIST and CIFAR and 70 epochs on CelebA. We use a mini-batch size of 100 and pad MNIST digits with zeros to make the size $32\times32$.

We use the official train, validation and test splits of CelebA. For MNIST and CIFAR, we set aside 10k train samples for validation. For random sample evaluation, we draw samples from $\mathcal{N}(0, I)$ for VAE and WAE-MMD and for all remaining models, samples are drawn from a multivariate Gaussian whose parameters (mean and covariance) are estimated using training set embeddings. For the GMM density estimation, we also utilize the training set embeddings for fitting and validation set embeddings to verify that GMM models are not over fitting to training embeddings. However, due to the very low number of mixture components (10), we did not encounter overfitting at this step. The GMM parameters are estimated by running EM for at most 100 iterations.

| | MNIST | CIFAR_10 | CELEBA |
|---|---|---|---|
| ENCODER: | $x \in \mathcal{R}^{32\times32}$ | $x \in \mathcal{R}^{32\times32}$ | $x \in \mathcal{R}^{64\times64}$ |
| | $\rightarrow$ CONV$_{128}$ $\rightarrow$ BN $\rightarrow$ RELU | $\rightarrow$ CONV$_{128}$ $\rightarrow$ BN $\rightarrow$ RELU | $\rightarrow$ CONV$_{128}$ $\rightarrow$ BN $\rightarrow$ RELU |
| | $\rightarrow$ CONV$_{256}$ $\rightarrow$ BN $\rightarrow$ RELU | $\rightarrow$ CONV$_{256}$ $\rightarrow$ BN $\rightarrow$ RELU | $\rightarrow$ CONV$_{256}$ $\rightarrow$ BN $\rightarrow$ RELU |
| | $\rightarrow$ CONV$_{512}$ $\rightarrow$ BN $\rightarrow$ RELU | $\rightarrow$ CONV$_{512}$ $\rightarrow$ BN $\rightarrow$ RELU | $\rightarrow$ CONV$_{512}$ $\rightarrow$ BN $\rightarrow$ RELU |
| | $\rightarrow$ CONV$_{1024}$ $\rightarrow$ BN $\rightarrow$ RELU | $\rightarrow$ CONV$_{1024}$ $\rightarrow$ BN $\rightarrow$ RELU | $\rightarrow$ CONV$_{1024}$ $\rightarrow$ BN $\rightarrow$ RELU |
| | $\rightarrow$ FLATTEN $\rightarrow$ FC$_{16\times M}$ | $\rightarrow$ FLATTEN $\rightarrow$ FC$_{128\times M}$ | $\rightarrow$ FLATTEN $\rightarrow$ FC$_{64\times M}$ |
| DECODER: | $z \in \mathcal{R}^{16} \rightarrow$ FC$_{8\times8\times1024}$ | $z \in \mathcal{R}^{128} \rightarrow$ FC$_{8\times8\times1024}$ | $z \in \mathcal{R}^{64} \rightarrow$ FC$_{8\times8\times1024}$ |
| | $\rightarrow$ BN $\rightarrow$ RELU | $\rightarrow$ BN $\rightarrow$ RELU | $\rightarrow$ BN $\rightarrow$ RELU |
| | $\rightarrow$ CONVT$_{512}$ $\rightarrow$ BN $\rightarrow$ RELU | $\rightarrow$ CONVT$_{512}$ $\rightarrow$ BN $\rightarrow$ RELU | $\rightarrow$ CONVT$_{512}$ $\rightarrow$ BN $\rightarrow$ RELU |
| | $\rightarrow$ CONVT$_{256}$ $\rightarrow$ BN $\rightarrow$ RELU | $\rightarrow$ CONVT$_{256}$ $\rightarrow$ BN $\rightarrow$ RELU | $\rightarrow$ CONVT$_{256}$ $\rightarrow$ BN $\rightarrow$ RELU |
| | $\rightarrow$ CONVT$_1$ | $\rightarrow$ CONVT$_1$ | $\rightarrow$ CONVT$_{128}$ $\rightarrow$ BN $\rightarrow$ RELU |
| | | | $\rightarrow$ CONVT$_1$ |

Conv$_n$ represents a convolutional layer with $n$ filters. All convolutions Conv$_n$ and transposed convolutions ConvT$_n$ have a filter size of $4\times4$ for MNIST and CIFAR-10 and $5\times5$ for CELEBA. They

all have a stride of size 2 except for the last convolutional layer in the decoder. Finally, $M = 1$ for all models except for the VAE which has $M = 2$ as the encoder has to produce both mean and variance for each input.

## D    EVALUATION SETUP

We compute the FID of the reconstructions of random validation samples against the test set to evaluate reconstruction quality. For evaluating generative modeling capabilities, we compute the FID between the test data and randomly drawn samples from a single Gaussian that is either the isotropic $p(\mathbf{z})$ fixed for VAEs and WAEs, a learned second stage VAE for 2sVAEs, or a single Gaussian fit to $q_\delta(\mathbf{z})$ for CV-VAEs and RAEs. For all models, we also evaluate random samples from a 10-component Gaussian Mixture model (GMM) fit to $q_\delta(\mathbf{z})$. Using only 10 components prevents us from overfitting (which would indeed give good FIDs when compared with the test set)[3].

For interpolations, we report the FID for the furthest interpolation points resulted by applying spherical interpolation to randomly selected validation reconstruction pairs.

We use 10k samples for all FID and PRD evaluations. Scores for random samples are evaluated against the test set. Reconstruction scores are computed from validation set reconstructions against the respective test set. Interpolation scores are computed by interpolating latent codes of a pair of randomly chosen validation embeddings vs test set samples. The visualized interpolation samples are interpolations between two randomly chosen test set images.

## E    EVALUATION BY PRECISION AND RECALL

|          | MNIST | | CIFAR-10 | | CELEBA | |
|----------|-------|-------|-------|-------|-------|-------|
|          | $\mathcal{N}$ | GMM | $\mathcal{N}$ | GMM | $\mathcal{N}$ | GMM |
| VAE      | 0.96 / 0.92 | 0.95 / 0.96 | 0.25 / 0.55 | 0.37 / 0.56 | 0.54 / 0.66 | 0.50 / 0.66 |
| CV-VAE   | 0.84 / 0.73 | 0.96 / 0.89 | 0.31 / 0.64 | 0.42 / 0.68 | 0.25 / 0.43 | 0.32 / 0.55 |
| WAE      | 0.93 / 0.88 | **0.98** / 0.95 | 0.38 / 0.68 | 0.51 / **0.81** | 0.59 / 0.68 | **0.69 / 0.77** |
| RAE-GP   | 0.93 / 0.87 | 0.97 / **0.98** | 0.36 / 0.70 | 0.46 / 0.77 | 0.38 / 0.55 | 0.44 / 0.67 |
| RAE-L2   | 0.92 / 0.87 | **0.98 / 0.98** | 0.41 / 0.77 | **0.57 / 0.81** | 0.36 / 0.64 | 0.44 / 0.65 |
| RAE-SN   | 0.89 / 0.95 | **0.98** / 0.97 | 0.36 / 0.73 | 0.52 / **0.81** | 0.54 / 0.68 | 0.55 / 0.74 |
| RAE      | 0.92 / 0.85 | **0.98 / 0.98** | 0.45 / 0.73 | 0.53 / 0.80 | 0.46 / 0.59 | 0.52 / 0.69 |
| AE       | 0.90 / 0.90 | **0.98** / 0.97 | 0.37 / 0.73 | 0.50 / 0.80 | 0.45 / 0.66 | 0.47 / 0.71 |

Table 2: Evaluation of random sample quality by precision / recall (Sajjadi et al., 2018) (higher numbers are better, best value for each dataset in bold). It is notable that the proposed ex-post density estimation improves not only precision, but also recall throughout the experiment. For example, WAE seems to have a comparably low recall of only $0.88$ on MNIST which is raised considerably to $0.95$ by fitting a GMM. In all cases, GMM gives the best results. Another interesting point is the low precision but high recall of all models on CIFAR-10 – this is also visible upon inspection of the samples in Fig. 9.

---

[3]We note that fitting GMMs with up to 100 components only improved results marginally. Additionally, we provide nearest-neighbours from the training set in Appendix G to show that our models are not overfitting.

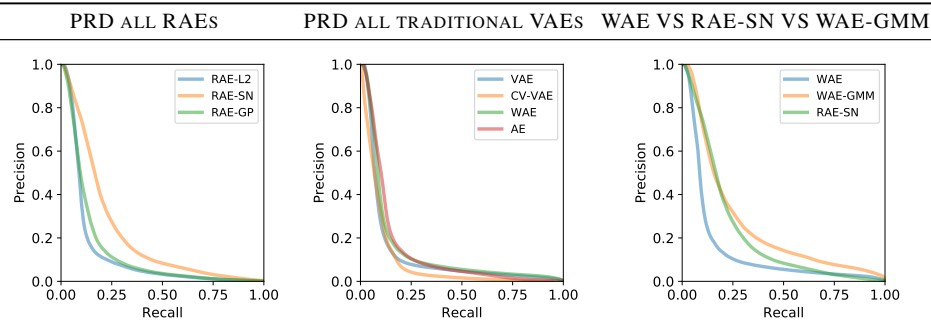

Figure 4: PRD curves of all RAE methods (left), reflects a similar story as FID scores do. RAE-SN seems to perform the best in both precision and recall metric. PRD curves of all traditional VAE variants (middle). Similar to the conclusion predicted by FID scores there are no clear winner. PRD curves for the WAE (with isotropic Gaussian prior) , WAE-GMM model with ex-post density estimation by a 10-component GMM and RAE+SN-GMM (right). This finer grained view shows how the WAE-GMM scores higher recall but lower precision than a RAE+SN-GMM while scoring comparable FID scores. Note that ex-post density estimation greatly boosts the WAE model in both PRD and FID scores.

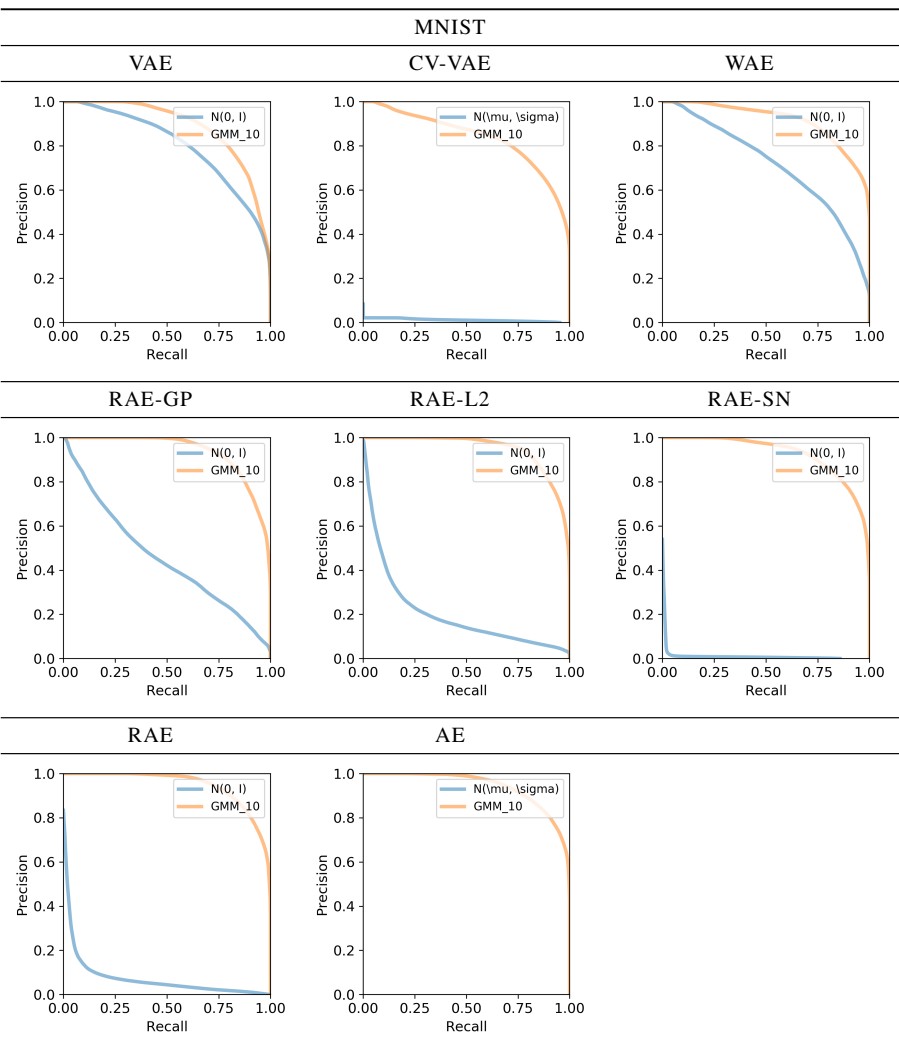

Figure 5: PRD curves of all methods on image data experiments on MNIST. For each plot, we show the PRD curve when applying the fixed or the fitted one by ex-post density estimation (XPDE). XPDE greatly boosts both precision and recall for all models.

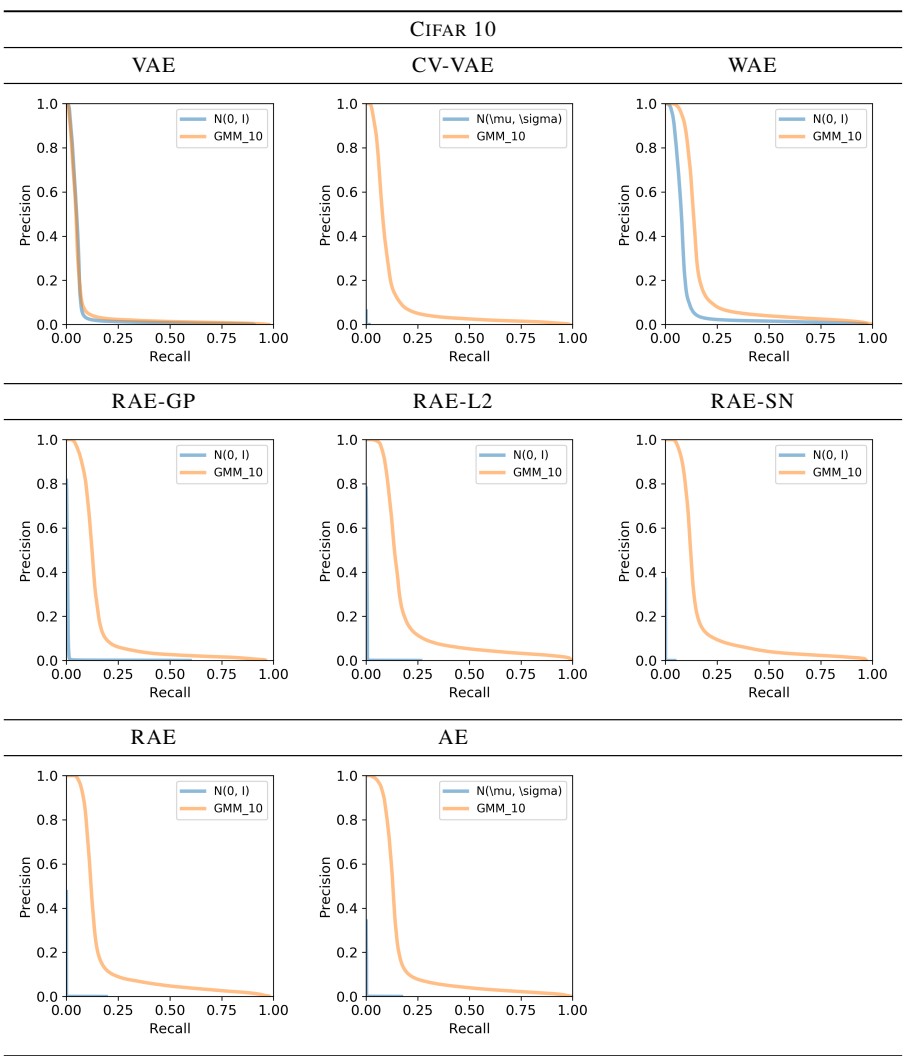

Figure 6: PRD curves of all methods on image data experiments on CIFAR10. For each plot, we show the PRD curve when applying the fixed or the fitted one by ex-post density estimation (XPDE). XPDE greatly boosts both precision and recall for all models.

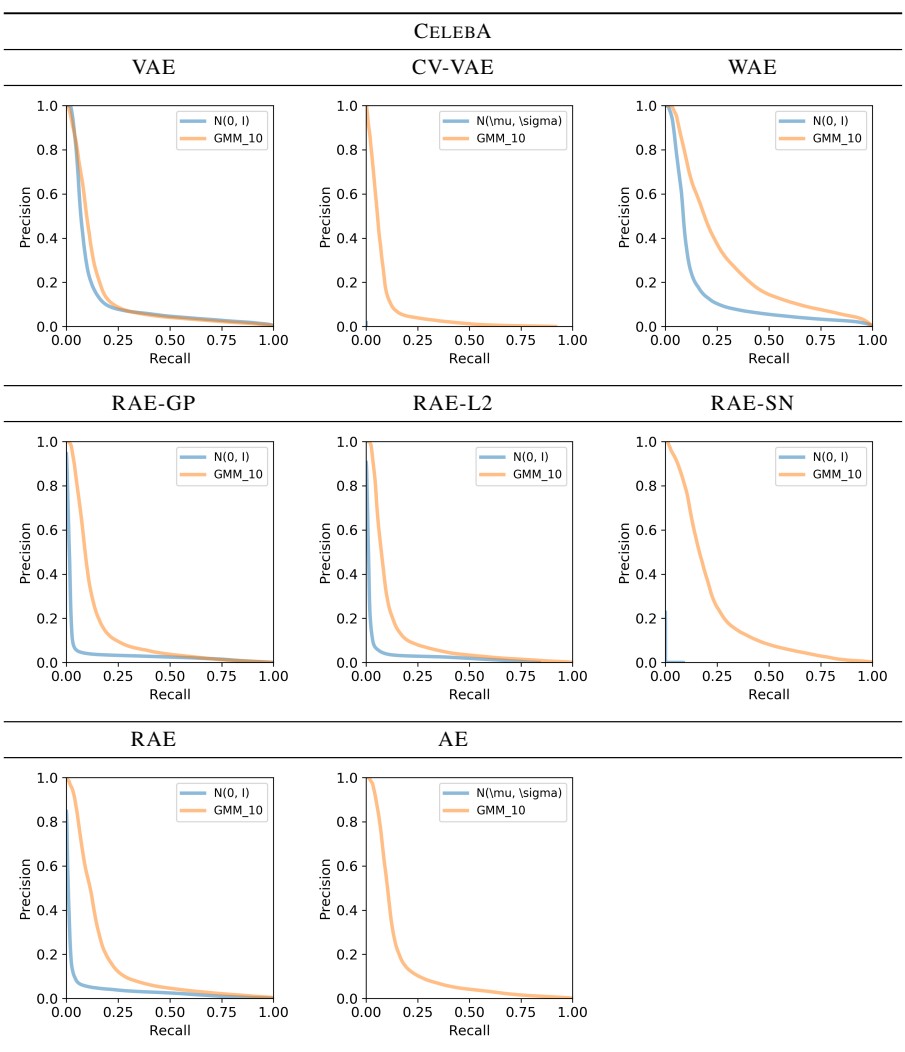

Figure 7: PRD curves of all methods on image data experiments on CELEBA. For each plot, we show the PRD curve when applying the fixed or the fitted one by ex-post density estimation (XPDE). XPDE greatly boosts both precision and recall for all models.

## F   MORE QUALITATIVE RESULTS

Figure 8: Qualitative evaluation for sample quality for VAEs, WAEs and RAEs on MNIST. Left: reconstructed samples (top row is ground truth). Middle: randomly generated samples. Right: spherical interpolations between two images (first and last column).

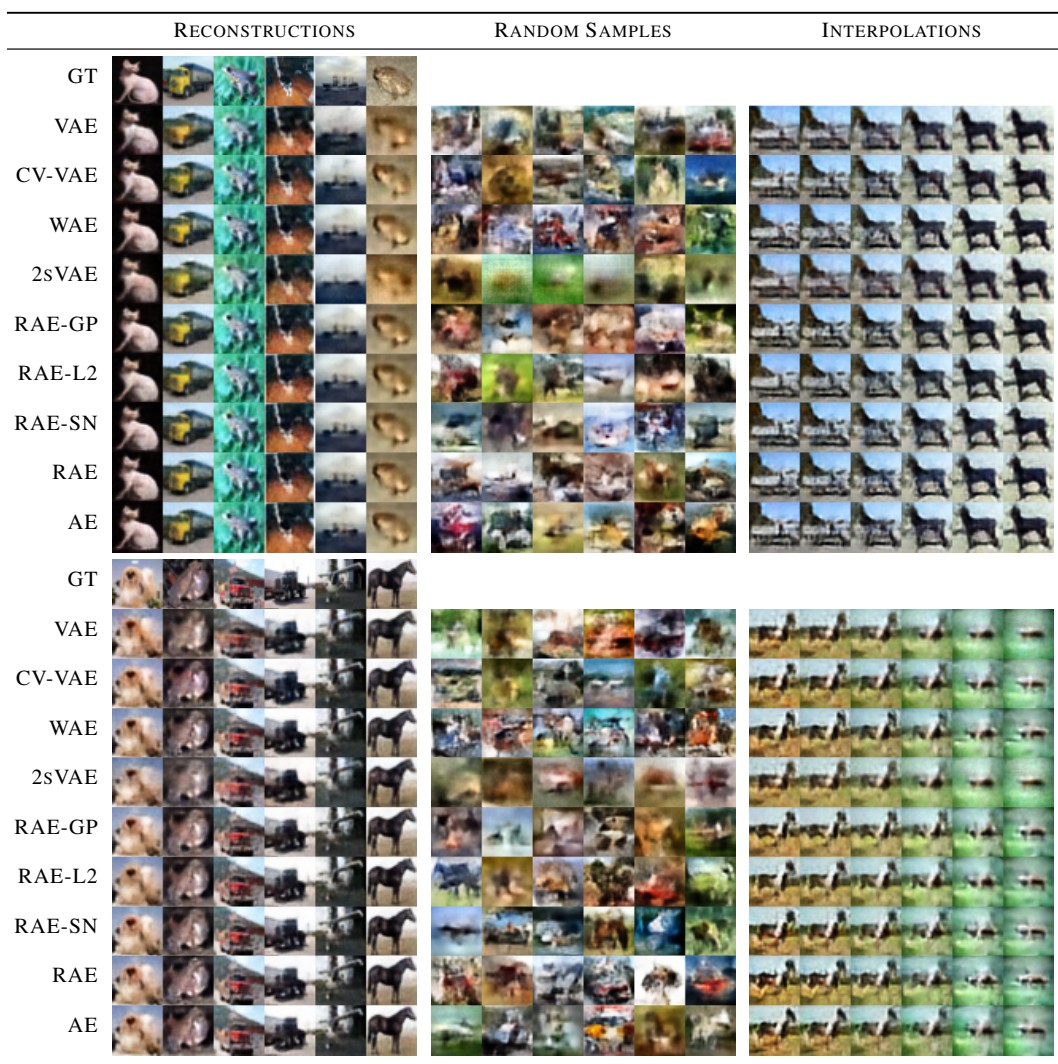

Figure 9: Qualitative evaluation for sample quality for VAEs, WAEs and RAEs on CIFAR-10. Left: reconstructed samples (top row is ground truth). Middle: randomly generated samples. Right: spherical interpolations between two images (first and last column).

# G INVESTIGATING OVERFITTING

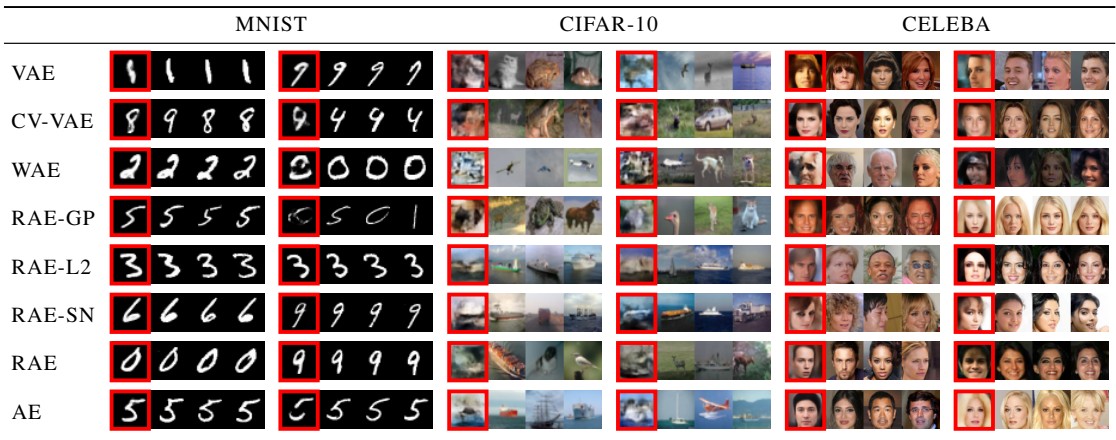

Figure 10: Nearest neighbors to generated samples (leftmost image, red box) from training set. It seems that the models have generalized well and fitting only 10 Gaussians to the latent space prevents overfitting.

## H    VISUALIZING EX-POST DENSITY ESTIMATION

To visualize that ex-post density estimation does indeed help reduce the mismatch between the aggregated posterior and the prior we train a VAE on the MNIST dataset whose latent space is 2 dimensional. The unique advantage of this setting is that one can simply visualize the density of test sample in the latent space by plotting them as a scatterplot. As it can be seen from figure 11, an expressive density estimator effectively fixes the miss-match and this as reported earlier results in better sample quality.

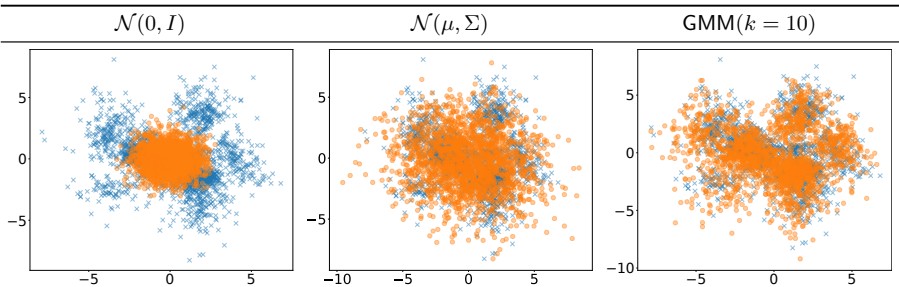

Figure 11: Different density estimations of the 2-dimensional latent space of a VAE learned on MNIST. The blue points are 2000 test set samples while the orange ones are drawn from the estimator indicated in each column: isotropic Gaussian (left), multivariate Gaussian with mean and covariance estimated on the training set (center) and a 10-component GMM (right). This clearly shows the aggregated posterior mismatch w.r.t. to the isotropic Gaussian prior imposed by VAEs and how ex-post density estimation can help fix the estimate.

Here in figure 12 we perform the same visualization on with all the models trained on the MNIST dataset as employed on our large evaluation in Table 1. Clearly every model depicts rather large mismatch between aggregate posterior and prior. Once again the advantage of ex-post density estimate is clearly visible.

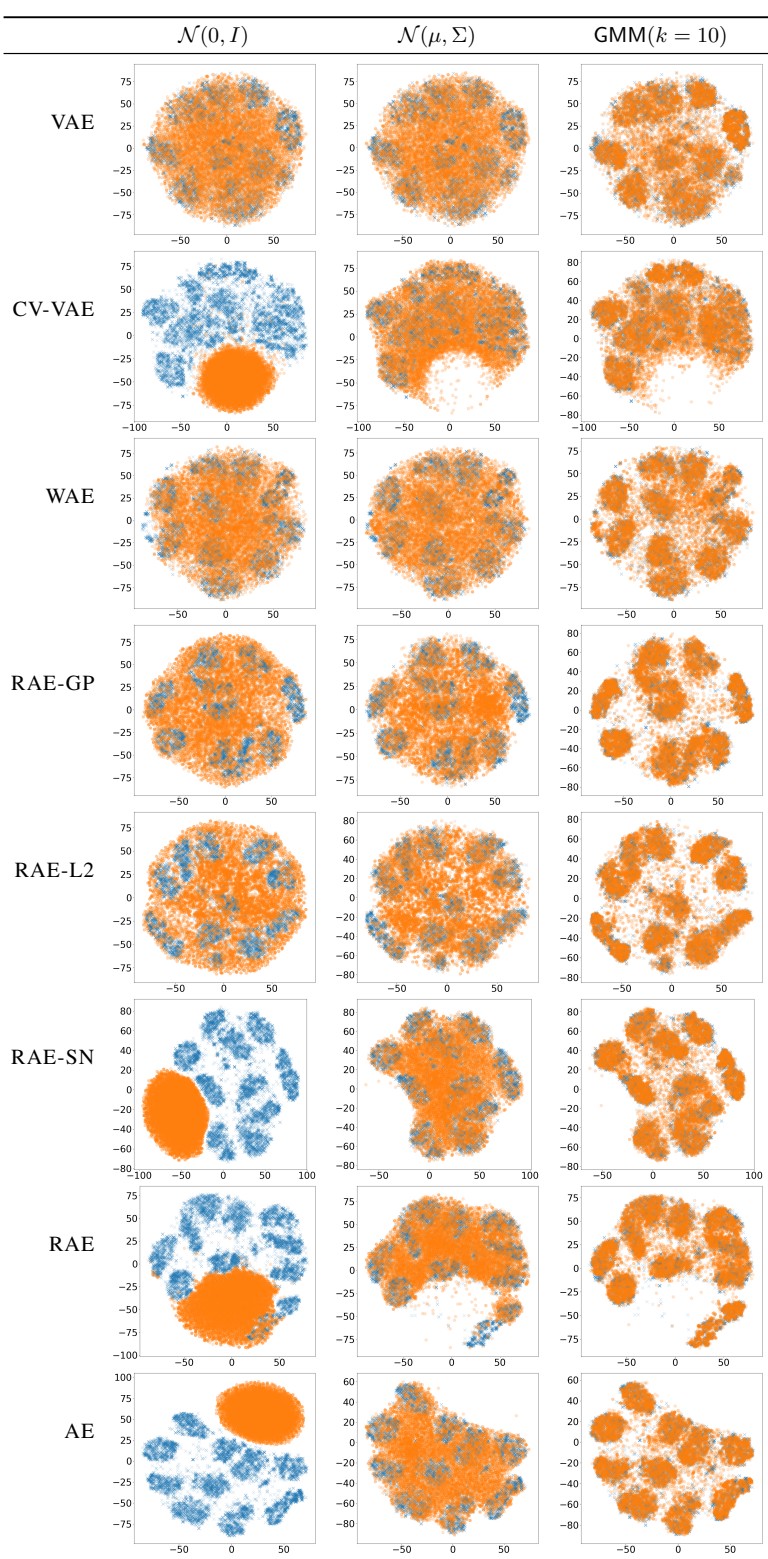

Figure 12: Different density estimations of the 16-dimensional latent spaces learned by all models on MNIST (see Table 1) here projected in 2d via T-SNE. The blue points are 2000 test set samples while the orange ones are drawn from the estimator indicated in each column: isotropic Gaussian (left), multivariate Gaussian with mean and covariance estimated on the training set (center) and a 10-component GMM (right). Ex-post density estimation greatly improves sampling the latent space.

|  | MNIST | | | | CIFAR | | | | CELEBA | | | |
|---|---|---|---|---|---|---|---|---|---|---|---|---|
|  | REC. | SAMPLES | | | REC. | SAMPLES | | | REC. | SAMPLES | | |
|  |  | $\mathcal{N}$ | GMM | Interp. |  | $\mathcal{N}$ | GMM | Interp. |  | $\mathcal{N}$ | GMM | Interp. |
| RAE-GP | 14.04 | **22.21** | 11.54 | 15.32 | 32.17 | 83.05 | 76.33 | 64.08 | **39.71** | 116.30 | 45.63 | 47.00 |
| RAE-L2 | 10.53 | 22.22 | **8.69** | **14.54** | 32.24 | **80.80** | 74.16 | 62.54 | 43.52 | 51.13 | 47.97 | 45.98 |
| RAE-SN | 15.65 | 19.67 | 11.74 | 15.15 | 27.61 | 84.25 | 75.30 | 63.62 | 36.01 | **44.74** | **40.95** | **39.53** |
| RAE | 11.67 | 23.92 | 9.81 | 14.67 | 29.05 | 83.87 | 76.28 | 63.27 | 40.18 | 48.20 | 44.68 | 43.67 |
| AE | 12.95 | 58.73 | 10.66 | 17.12 | 30.52 | 84.74 | 76.47 | **61.57** | 40.79 | 127.85 | 45.10 | 50.94 |
| AE-L2 | 11.19 | 315.15 | 9.36 | 17.15 | 34.35 | 247.48 | 75.40 | 61.09 | 44.72 | 346.29 | 48.42 | 56.16 |
| RAE-GP-L2 | **9.70** | 72.64 | 9.07 | 16.07 | 33.25 | 187.07 | 79.03 | 62.48 | 47.06 | 72.09 | 51.55 | 50.28 |
| RAE-L2-SN | 10.67 | 50.63 | 9.42 | 15.73 | **24.17** | 240.27 | **74.10** | 61.71 | 39.90 | 180.39 | 44.39 | 42.97 |
| RAE-SN-GP | 17.00 | 139.61 | 13.12 | 16.62 | 33.04 | 284.36 | 75.23 | 62.86 | 63.75 | 299.69 | 71.05 | 68.87 |
| RAE-L2-SN-GP | 16.75 | 144.51 | 13.93 | 16.75 | 29.96 | 290.34 | 74.22 | 61.93 | 68.86 | 318.67 | 75.04 | 74.29 |

Table 3: Comparing multiple regularization schemes for RAE models. The improvement in reconstruction, random sample quality and interpolated test samples is generally comparable, but hardly much better. This can be explained with the fact that the additional regularization losses make tuning their hyperparameters more difficult, in practice.

# I  COMBINING MULTIPLE REGULARIZATION TERMS

The rather intriguing facts that AE without explicit decoder regularization performs reasonably well as seen from table 1, indicates that convolutional neural networks when combined with gradient based optimizers inherit some implicit regularization. This motivates us to investigate a few different combinations of regularizations e.g. we regularize the decoder of an auto-encoder while drop the regularization in the $z$ space. The results of this experiment is reported in the row marked AE-L2 in table 3.

Further more a recent GAN literature **?** report that often a combination of regularizations boost performance of neural networks. Following this, we combine multiple regularization techniques in out framework. However note that this rather drastically increases the hyper parameters and the models become harder to train and goes against the core theme of this work, which strives for simplicity. Hence we perform simplistic effort to tune all the hyper parameters to see if this can provide boost in the performance, which seem not to be the case. These experiments are summarized in the second half of the table 3

