# OpenReview forum: "From Variational to Deterministic Autoencoders"
_ICLR.cc/2020/Conference — Accept (Poster)_

### Official Review · AnonReviewer2 · 2019-10-21
**Official Blind Review #2**

**Rating:** 6

**Review:**

This paper propose an extension to deterministic autoencoders. Motivated from VAEs, the authors propose RAEs, which replace the noise injection in the encoders of VAEs with an explicit regularization term on the latent representations. As a result, the model becomes a deterministic autoencoder with a L_2 regularization on the latent representation z. To make the model generalize well, the authors also add a decoder regularization term L_REG. In addition, due to the encoder in RAE is deterministic, the authors propose several ex-post density estimation techniques for generating samples.

The idea of transferring the variational to deterministic autoencoders is interesting. Also, this paper is well-written and easy to understand. However, in my opinion, this paper needs to consider more cases for autoencoders and needs more rigorous empirical and theoretical study before it can be accepted. Details are as follow:

1. The RAEs are motivated from VAEs, or actually CV-VAEs as in this paper. More precisely, the authors focus on VAEs with a constant covariance Gaussian distribution as the variational distribution and a Gaussian distribution with the identity matrix as the covariance matrix as the model likelihood. However, there might be many other settings for VAEs. For example, the model likelihood can be a Gaussian distribution with non-constant covariance, or even some other distributions (e.g. Multinomial, Bernoulli, etc). Similarly, the variational distribution can be a Gaussian distribution with non-constant covariance, or even some more complicated distributions that do not follow the mean-field assumption. Any of these more complex models may not be easily transferred to the RAE models that are mentioned in this paper. Perhaps it is better if the authors can consider RAEs for some more general VAE settings.

2. Perhaps the authors needs more empirical study, especially on the gain of RAE over CV-VAE and AE.
a) As the motivated model (CV-VAE) and the most related model in the objective (AE), they are not appearing in the structured input experiment (Section 6.2). It will be great if they can be compared with in this experiment.
b) The authors did not show us clearly whether the performance gain of RAE over VAE, AE and CV-VAE is due to the regularization on z (the term L_z^RAE) or the decoder regularization (the term L_REG) in the experiments. In table 1, the authors only compare the standard RAE with RAE without decoder regularization, but did not compare with RAE without the regularization on z (i.e. equivalent to AE + decoder regularization) and CV-VAE + decoder regularization. The authors would like to show that the explicit regularization on z is better than injecting the noise, hence the decoder regularization term should appear also in the baseline methods. It is totally possible that perhaps AE + decoder regularization or CV-VAE + decoder regularization perform better than RAE.
c) The authors did not show how they tune the parameter \sigma for CV-VAE. Since the parameter \beta in the objective of RAE is tunable, for fair comparison, the authors needs to find the best \sigma for CV-VAE in order to get the conclusion that explicit regularization is better than CV-VAE.
d) Although the authors mention that the 3 regularization techniques perform similarly, from Table 1, it is still hard to decide which one should we use in practice in order to get a performance at least not too much worse compared to the baseline methods. RAE-GP and RAE-L2 perform not well on CelebA while RAE-SN perform not well on MNIST, compared to the baseline methods. We know that the best performance over the 3 methods is always comparable to or better than the baselines, but not none of the single methods do. It is better if the authors can provide more suggestions on the choice for decoder regularization for different datasets.

3. The authors provided a theoretical derivation for the objective L_RAE (Equation 11), but this is only for the L_GP regularization. Besides, this derivation (in Appendix B) has multiple technique issues. For example, in the constraints in Equation 12, the authors wrote ||D_\theta(z1) - D_\theta(z2)|| < epsilon for all z1, z2 ~ q_\phi(z | x), this is impossible for CV-VAE since this constraint requires D_theta() to be bounded while q_\phi(z | x) in CV-VAE has an unbounded domain. Moreover, in the part  (||D_\theta(z1) - D_\theta(z2)||_p=\nabla D_\theta(\tilde z)\cdot ||z_1-z_2||_p) of Equation 13, \nabla D_\theta(\tilde z) is a vector well the other two terms are scalars, which does not make sense. There are many other issues as well. Please go through the proof again and solve these issues.


Questions and additional feedback:

1. Can the authors provide more intuitions why do you think the explicit regularization works better compared to the noise injection? Can you provide a theoretical analysis on that?

2. Can the authors provide some additional experiments as mentioned above? Also, can the authors provide more details about how do they tune the parameters \beta and \lambda?

========================================================================================================

After the rebuttal:

Thanks the authors for the detailed response and the additional experiments. I agree that the additional experiment results help to support the claims from the authors, especially for the CV-VAE for the structured data experiments and the AE + L2 experiment. So I think now the authors have more facts to support that RAE is performing better compared to the baseline methods.

Therefore, I agree that after the revision, the proposed method RAE is supported better empirically. So I am changing my score from "weak reject" to "weak accept". But I still think the baseline CV-VAE + regularization is important for Table 1 and the technical issues in the theoretical analysis needs to be solved. Hope the authors can edit them in the later version.

**Experience Assessment:**

I have published one or two papers in this area.

**Review Assessment: Checking Correctness Of Derivations And Theory:**

I carefully checked the derivations and theory.

**Review Assessment: Checking Correctness Of Experiments:**

I carefully checked the experiments.

**Review Assessment: Thoroughness In Paper Reading:**

I read the paper thoroughly.

---

> ### Author Response · Authors · 2019-11-07
> **A valuable stepping stone towards alternative generative modeling**
>
> Q1) <RAE compared against all VAE settings> We agree that having a deterministic framework that is able to extend and perform better than *any possible VAE variant* would be highly relevant and convenient. However, we believe that even if our paper does not achieve this ambitious goal, it has great value. Indeed, we introduce RAEs for a subset of VAEs parametrizations, we demonstrate their practical effectiveness as replacements for generative modeling and pave the way to future interesting extensions.
>
> We start our analogy on the Gaussian VAE with isotropic prior and constant variance. Therefore, the natural competitors of our RAE framework are the Gaussian VAEs. As argued in Section 2, this is arguably still the most commonly employed VAE for generative modeling (partly because more complex variants are provably tougher to train, cf. our discussion in Section 2).
> Furthermore, please note that i) we extend our empirical comparison also to WAEs, 2-stage VAEs and GVAEs and ii) some trivial change in the distributional assumptions over the VAE decoder can be easily ported in the RAE framework, e.g., a Bernoulli distribution would require to change the MSE to a CE loss.
>
>
> Q2) <More experiments> We agree that more experiments can be helpful in strengthening the outcomes of our "autopsy". Unfortunately, the possible space of all experimental setting combinations is exponential. We believe that our experimental design is adequately devised to highlight the effectiveness of our framework, and most importantly in a rigorous way. We will explain how and why in the following points. Nevertheless, we remark that we are open to run additional experiments during the rebuttal.
>
> a) <CV-VAE for SOP> Please note that the structure output prediction (SOP) experiments are circa one order of magnitude longer to run, given the repetitions of the costly BO. This is why we selected a single RAE scheme, and we selected the fastest one with L_L2 reg. We are in the process of running one additional case involving the CV-VAE as requested.
>
> b) <Effect of L_Z> In our experimental setting one can quantify the impact of introducing the L_Z term by looking at experiments involving AEs (not adding L_Z in their objective) and vanilla RAEs (including only L_Z). AEs achieve comparable, or only slightly less, random sample FIDs. Time permitting, we will run an additional experiment with only AE + explicit regularization.  We remark that the outcome of these additional experiments are not invalidating our claim: implicit or explicit decoder regularization on a deterministic autoencoder deliver a generative model competitive with VAEs and variants.
>
> c) <tuning hyperparameter>  We perform the same grid search by binary search over some reasonable interval for all hyperparameters involved and spent a reasonably equal amount of time for all models involved in our comparison. See also our answer to Reviewer#1.
>
> d) <which RAE?> Indeed, there is no clear winner, overall. As stated in the paper, we recommend RAE-L2 for its simplicity. We adopted it for the more involved SOP experiments, delivering far better results than GVAEs. Furthermore, we would like to highlight that use of different types of regularization emphasizes the point that as predicted by the theory a regularization is necessary and any implicit or explicit one works well in practice.
>
> Q3) <Derivation only for L_GP> We agree with the reviewer that providing a general theoretical derivation for every possible regularization schemes would be very sensible. As it is highly non-trivial and impactful, it could potentially be worth a best paper award. However, we believe this perspective should not diminish the value of our contribution. We point out that, while L_GP and L_SN are introduced more as heuristics from the GAN literature, here we provide a sound theoretical derivation for L_GP in the autoencoder framework. Furthermore, the L_L2 regularization is theoretically backed by the Tikhonov regularization framework for general function approximators.
>
> Q4) <unbounded domain> We appreciate and thank that the reviewer for providing such a thorough check of our proof. As we mention after eq.13, we are interested in q_{\phi}(z | x) with bounded support, e.g., a uniform distribution over a hypersphere in z. We will specify this before eq. 13, to make the assumption more clear as suggested.
>
> <dimension mismatch> We thank the reviewer for having spotted this typo! Indeed, we are missing the p-norm over the (vector) gradient, which appears on the right side of same equation. This does not hinder in any way our proof (the right side of the equation is correct as the remaining part). As suggested, we will thoroughly proofread the derivation and update the manuscript accordingly.

---

> ### Author Response · Authors · 2019-11-13
> **Additional experiments and evidence to support our claims**
>
> Q2a) As requested, we run the CV-VAE variants for the structured data experiments, which we named GCVVAE. Please see the new revision we uploaded, we reported results in Figure 2 in our experimental section. As one can see from there,  making the variance constant as is the case with CVVAEs, provides a boost in the expression and molecule datasets when compared to the GVAE. While no significant difference is perceivable w.r.t. a GRAE for expressions, on the more challenging molecules datasets, RAE still deliver better average scores and more significant molecules (while the number of valid molecules is slightly better for the CVVAE, even though not significantly, given the std).
>
> Q2b) Furthermore, we run as requested the AE model plus a regularization term, on the image data. We employed the L_L2 regularization for all datasets and with all the different XPDE estimates. We reported results in Table 1 in our original experimental setting. It is visible from there that by not constraining the Zs during optimization results in a latent space that is highly non-isotropic-Gaussian. Fortunately, XPDE with GMMs helps deliver better samples by fixing this mismatch. Interpolations are only very slightly worse, but still comparable with other models. As expected, this is due that the latent codes to be allowed to "strech" as much as they want. As in our claims, regularization on the decoder helps the interpolations to still deliver very reasonable samples.
>
> We will report additional combinations of AE+REG in a new revision of the paper.
>
> We welcome any additional suggestion from the reviewer to strengthen our claims empirically.

---

### Official Review · AnonReviewer1 · 2019-10-22
**Official Blind Review #1**

**Rating:** 8

**Review:**

The paper analyzes Variational Autoencoders and formulates an alternative perspective on many common problems in the VAE framework like a mismatch between the aggregated posterior and the marginal distribution over latent variables. The authors perform an autopsy of the VAE objective and show how it could be formulated as a constrained optimization problem. Interestingly, the paper proposes to remove all stochasticity from an encoder and a decoder and use different regularizers instead. In order to be able to sample from the model, the marginal distribution over latents is trained after the encoder and decoder are learned.  In general, I find the paper very interesting and important for the VAE community.

- (An open question) The marginal distribution p(z) allows to turn the AE into a generative model. However, it also serves as a code model in a compression framework (i.e., it is used in the adaptive entropy coding). It would be interesting to see how the proposed approach would compare to other compression methods.

- It would be interesting to see how the post-training of p(z) performs. Maybe it would be a good idea to run some toyish problem (e.g., dim(z)=2 on MNIST) and see how the GMM fits samples from the aggregated posterior.

- Did the authors try a mix of different regularizers they proposed to use? For instance, L2 + SN?

- Could the authors comment on the choice of hyperparameters (weights in the objective)?

======== AFTER REBUTTAL ========
I would like to thank the authors for answering my questions. I really appreciate all effort to provide more empirical evidence. I still believe that this paper is extremely important for the AE/VAE community and it sheds some new light on representation learning. In my opinion the paper should be accepted. Therefore, I decided to rise my score from "Weak Accept" to "Accept".

**Experience Assessment:**

I have published in this field for several years.

**Review Assessment: Checking Correctness Of Derivations And Theory:**

I carefully checked the derivations and theory.

**Review Assessment: Checking Correctness Of Experiments:**

I assessed the sensibility of the experiments.

**Review Assessment: Thoroughness In Paper Reading:**

I read the paper thoroughly.

---

> ### Author Response · Authors · 2019-11-07
> **Interesting experimental suggestions**
>
> Q1) <relation to compression> We thank the reviewer for this very interesting perspective. We are not experts in adaptive entropy coding, but we will try to assess the meaningfulness of the learned codes by some down-stream task metric. For instance, one could rank the Z codes learned without supervision by RAEs and VAEs on MNIST by the accuracy scored by a logistic regressor.
>     We welcome further suggestions and pointers to other similar settings to augment our experimental design.
>
> Q2) <visual intuition on XPDE> This is a valuable suggestion and we are in the process of running the suggested 2D visual comparison. We will update the paper accordingly. Moreover, we can extend this visual inspection to the higher dimensional cases by showing a 2D-projection, e.g., via t-sne, provided they are meaningful.
>
> Q3) <mixing regularizers>  As recent works in the GAN literature have shown, combining different regularizers might lead to better results (e.g., cf. [1], [2]).  We therefore agree that this is a valuable direction. In our experimental design we strove for rigor and, most importantly, for simplicity: our aim is to prove that even a single explicit regularization scheme (or no explicit like for AE) can deliver a smooth latent space and high quality samples. Furthermore, combining more penalties would make training a RAE less practical: more hyperparameters to be tuned and longer training time.
>     Nevertheless, we will add the suggested experiments on some regularizer combinations to the manuscript as soon as we have the results.
>
> Q4) <choice of hyperparameters> We optimized all hyperparameters for each model separately on each dataset, by optimizing the random sample FID on a held-out set. To this aim, we executed a grid search by performing a binary search starting from some common intervals. We will provide all details to reproduce this search in the Appendix. Note that we reserved the same effort to tune the hyperparameters of all models, achieving better FIDs for our competitors w.r.t. other papers (e.g. our VAEs generate better samples than reported in the original WAE paper).
>
> [1] - Lucic et al, "Are GANs Created Equal? A Large-Scale Study", NeurIPS 2018
> [2] - Kurach et al. "A Large-Scale Study on Regularization and Normalization in GANs", ICML 2019

---

> ### Author Response · Authors · 2019-11-13
> **Additional experiments and visualizations to provide insight**
>
> Q2) We provided the suggested visualization in Appendix H. There, we visualize how XPDE affects the estimation of the latent space on MNIST when employing either 2 dimensions or 16 as in the paper (for which we employ t-sne to project to a 2D plane).
>
> In both cases we plot the original test set samples against the random samples generated by the model when equipped with an isotropic Gaussian, or a Gaussian whose mean and covariance are estimated from data, or by employing a 10-component GMM.
> The positive effect of XPDE is clearly visualized: a better estimator over the latent space delivers samples that better match the original density. This is ultimately translated into better samples.
>
> Q3) We are still in the process of running additional experiments for combinations of regularizers. We will report them in the next revision of the paper.
>
> Please see our general comments on the updated revision. We welcome further suggestions to improve presentation and experiments.

---

> ### Author Response · Authors · 2019-11-15
> **All combinations of regularization schems**
>
> We added results for all possible combinations of the regularization schemes proposed on our RAEs in Appendix I.
>
> The   improvement in reconstruction, random sample quality and   interpolated test samples is generally comparable, but hardly much better.  This can be explained with the fact that the additional regularization losses make tuning their hyperparameters more difficult, in practice.
>
> As stated in the paper, we would recommend practitioners to use RAE+L2 for simplicity.

---

### Official Review · AnonReviewer3 · 2019-10-28
**Official Blind Review #3**

**Rating:** 6

**Review:**

The paper studies (the more conventional) deterministic auto-encoders, as they are easier to train than VAE. To then try to maintain the model's capability of approximating the data distribution and to draw/synthesize new unseen samples, the paper both looks at imposing additional regularization terms towards a smooth decoder and proposes to sample from a latent distribution that's induced from empirical embeddings (similar to an aggregate posterior in VAE). Experiments are mostly around contrasting VAEs with the proposed RAEs in terms of comparing the quality of the generated samples.

The paper is trying to answer an important and meaningful question, i.e. can a deterministic auto-encoder learn a meaningful latent space and approximates the data distribution just as well as the much more complicated VAEs. And they've made some interesting and reasonable trials. The methods developed in the paper are mostly easy to understand and also well motivated in general.
However my main concern is on the empirical studies, as they don't seem particularly convincing to justify the claimed success of the RAE over VAE. That said, the evaluation of generative models is by itself still an active research topic in development, which certainly adds to the difficulty of coming up with a sensible and rigorous comparative study in this regard.

Detailed comments and questions.
1. On "interpolation": it might provide some insights on how well the decoder can "generalize" to unseen inputs (which is still important), but otherwise seems to provide very little insight on how well the actual "distribution" is being approximated, as it's much more about "coverage" of the support of the distribution than the actual probabilities themselves?
2. I'm not so sure FID can be regarded as the golden standard when it comes to comparing data generation qualities, and especially for the given setting, as it might give all those methods employing the so-called "ex-post density estimation" an edge over VAEs due to the scraping of the gap between the aggregate posterior and the prior (by design).
3. From Table 2, according to the "Precision/Recall" criteria, WAE clearly out-performs RAE on the CelebA dataset, contradicting the result with FID in Table 1. I think this might need a closer look (as to why this happened) and certainly should be worth some discussions in the paper.

**Experience Assessment:**

I have read many papers in this area.

**Review Assessment: Checking Correctness Of Derivations And Theory:**

I assessed the sensibility of the derivations and theory.

**Review Assessment: Checking Correctness Of Experiments:**

I assessed the sensibility of the experiments.

**Review Assessment: Thoroughness In Paper Reading:**

I read the paper at least twice and used my best judgement in assessing the paper.

---

> ### Author Response · Authors · 2019-11-07
> **Remarks on our empirical evaluation**
>
> Q1) <meaningfulness of interpolation> We agree with the reviewer that interpolation experiments serve the purpose to prove that the learned latent space is smooth and contains no "holes". Note that RAEs do not define an explicit probability distribution p(X), therefore assessing the distance between the true data distribution can only be done through statistical tests on samples (e.g. FID, PRD).
>
> Q2) <FID golden standard> We agree with the reviewer that the evaluation of generative models is a nontrivial problem. Among many recent alternatives, FID is still the most popular criterion adopted to assess the distance between two distribution via their samples. We have reported PRD in the appendix but we are open to run any other metric the reviewer suggests.
>     <XPDE boosts FID> We agree that eX-Post Density Estimation (XPDE) boosts FID scores. This is done by design, since XPDE is introduced to fix the posterior mismatch. Note that we also apply XPDE to all VAE variants (see Table 1 under the GMM column) to demonstrate that it effectively reduces the gap between the prior and the aggregate posterior. This per se might be a highly relevant contribution for the generative modeling community and its simplicity might have let it pass unnoticed so far.
>
> Q3> <better PRD for WAEs> We are in the process or recomputing the PRD scores on WAE to assess if it is a typo or it depends on the sensitivity of PRD w.r.t. the clustering we adopted (in which case we will discuss this sensitivity issue in the paper).

---

> ### Author Response · Authors · 2019-11-13
> **Update on PRD scores and more experiments**
>
> Q1+Q3) We unraveled the WAE's FID scores "mystery". We reported values in Table 2 in Appendix E that are all subject to the same beta value for computing PRD [cf. Sajjadi M. et al. 2018]. These values flatten the understanding of the trade-off between precision and recall, which is otherwise perceivable by looking at their curves.
> Therefore, we have augmented Appendix E to now include the PRD curves for these models and added a discussion in Figure 4.
> Indeed, from Figure 4 it is visible how WAE+GMM achieves better recall, but much less precision overall than a RAE_SN+GMM. This trade-off is also not visible by the single FID score, where the WAE+GMM achieves something slightly less than RAEs+GMMs, and is also distorted while fixing a single beta, as we report in the Table.
>
> We will report all PRD curves for all models and datasets in the final revision of the paper, as well as extensively commenting on this caveat of the evaluation. We thank the reviewer for pointing this out.
>
> Furthermore, we have expanded our experimental setting as requested by other reviewers, please see our general comment on the updated revision.

---

> > ### Author Response · Authors · 2019-11-15
> > **More PRD curves**
> >
> > We uploaded all the PRD curves for all the models on all the image data experiments.
> > The positive effect of performing XPDE is clearly visible from these plots.

---

### Author Response · Authors · 2019-11-07
**General comments to all reviews**

We thank all the reviewers for appreciating the direction of our work and its relevance for the generative modeling community.
We agree that the evaluation of generative models is a nontrivial problem and a very active field.
We appreciate the constructive criticisms and would gladly accommodate additional experiments that would improve the paper's quality. To this end we would like to engage with every question individually and distill the core addendum to the paper.
However before that, we would like to emphasize that the scope of the paper is not to introduce a model that always supersedes all possible VAE variants and extensions. Instead, we aim to provide a simpler alternative to the most popular variants, that yields a as-good-as or even better generative mechanism. Most importantly, by doing so we intend to shed light over some common unquestioned assumptions and open a novel perspective in the current research landscape of generative modeling.

---

### Author Response · Authors · 2019-11-13
**New revision uploaded: more experiments and plots**

We thank again all the reviewers for the valuable feedback.

We have uploaded a new revision of the paper including the following changes:

+ we investigated the PRD values for the WAE on CelebA and provided a better insight of the phenomenon by visualizing the whole PRD curves (see Appendix E)

+ we built more intuition on XPDE and the aggregated posterior mismatch problem by visualizing the learned latent space and the fit by different estimators on 2D MNIST and the models employed in the experiments via t-sne projections (see Appendix H)

+ we have run the AE+L2 model for all image datasets and XPDE variants, reporting FIDs for reconstructions, random samples and interpolation (see Table 1)

+ we have extended the structured object experiments to the CVVAE, generating and scoring equations and molecules in Figure 2

Please refer to our individual comments for more details and do not hesitate to ask for clarifications and more experiments!

---

### Author Response · Authors · 2019-11-15
**New revision: more experiments**

We have uploaded a new revision of the paper including the following experiments and visualizations:

+ all PRD curves for all models on all image datasets, showing the positive impact of performing XPDE (Appendix E)

+ all possible combinations of the regularizers applied to RAEs (Appendix I)

---

### Decision · Program_Chairs · 2019-12-19

**Decision:**

Accept (Poster)

**Comment:**

This paper proposes an extension to deterministic autoencoders, namely instead of noise injection in the encoders of VAEs to use deterministic autoencoders with an explicit regularization term on the latent representations. While the reviewers agree that the paper studies an important question for the generative modeling community, the paper has been limited in terms of theoretical analysis and experimental validation. The authors, however, provided further experimental results to support the claims empirically during the discussion period and the reviewers agree that the paper is now acceptable for publication in ICLR-2020.